# Experimental Investigation and Parametric Optimization of the Tungsten Inert Gas Welding Process Parameters of Dissimilar Metals

**DOI:** 10.3390/ma15134426

**Published:** 2022-06-23

**Authors:** Anteneh Teferi Assefa, Gulam Mohammed Sayeed Ahmed, Sagr Alamri, Abhilash Edacherian, Moera Gutu Jiru, Vivek Pandey, Nazia Hossain

**Affiliations:** 1Department of Mechanical Engineering, Adama Science and Technology University, Adama 1888, Ethiopia; antenehteferi2020@gmail.com (A.T.A.); dursemoti2012@gmail.com (M.G.J.); vivek.pandey@astu.edu.et (V.P.); 2Center of Excellence (COE) for Advanced Manufacturing Engineering, Program of Mechanical Design and Manufacturing Engineering, School of Mechanical, Chemical and Materials Engineering, Adama Science and Technology University, Adama 1888, Ethiopia; 3Department of Mechanical Engineering, College of Engineering, King Khalid University, Abha 61421, Saudi Arabia; salamri@kku.edu.sa (S.A.); edalheriad@kku.edu.sa (A.E.); 4School of Engineering, RMIT University, Melbourne, VIC 3001, Australia; bristy808.nh@gmail.com

**Keywords:** dissimilar metal, gas tungsten arc welding, mechanical properties, multi-response optimization, microstructure properties

## Abstract

Special attention is required when joining two materials with distinct chemical, physical and thermal properties in order to make the joint bond robust and rigid. The goal of this study was to see how significantly different tungsten inert gas (TIG) welding process parameters (welding current, gas flow rate, root gap, and filler materials) affect mechanical properties (tensile, hardness, and flexural strength), as well as the bead width and microstructural properties, of dissimilar welds In comparison to SS 316 and AISI 1020 low-carbon steel. TIG welding parameters were optimized in this study using a Taguchi-based desirability function analysis (DFA). From the experimental results, it was observed that welded samples employing ER-309L filler wires had a microstructure consisting of a delta ferrite network in an austenite matrix. The tensile strength experimental results revealed that welding current, followed by GFR, was a highly influential parameter on tensile strength. Weld metals had higher hardness and flexural strength than stainless steel and carbon steel base metals. This was supported by the fact that the results of our tests had hardness ratings greater than a base for the FZ and HAZ, and that no crack was observed in the weld metal following U-shape flexural bending. Welding current has a significant impact on the bead width of welded specimens, followed by root gap. Furthermore, the dissimilar welded sample responses were optimized with a composite desirability percentage improvement of 22.90% by using a parametric setting of (A2B4C4D2). Finally, the validation of the experiment was validated by our confirmation test results, which agreed with the predictive optimum parameter settings.

## 1. Introduction

Manufacturers are focused on dissimilar materials joining to reduce manufacturing costs and build lightweight components. Steel structures are lighter and more cost effective when their structural components are made of different steels [1]. Chemical, petrochemical, nuclear, power generation, and other industries use a variety of dissimilar steel joints [2,3]. When welding of components, dissimilar joints are unavoidable. Joining dissimilar steels is typically more difficult than joining similar steels [4,5]; this is caused by a variety of causes, including changes in chemical composition and thermal expansion coefficients. Gas tungsten arc welding (GTAW), also known as tungsten inert gas welding (TIG), ref. [6] is an arc welding process that creates an electric arc using a non-consumable tungsten electrode. TIG welding provides outstanding welding with a coalescence of heat generated through an electric arc between a tungsten electrode and the steel [7,8]. TIG welding is a simple, fumeless, and spatter-free process that requires little or no finishing. This study will look at the effects of welding current, root gap, gas flow rate, and filler metal type on the welding of the dissimilar metals SS316 stainless steel and AISI 1020 mild steel. Mechanical properties, such as tensile strength, hardness strength, and flexural bending strength, and microstructural properties were used to assess the TIG welding process parameters. Finally, this study investigates and optimizes the effect of welding factors on welding quality.

## 2. Literature Review

### 2.1. Consideration during Welding Dissimilar Metals

Previous studies on the TIG welding of diverse and dissimilar metals can be split into three categories: characterization, parameter optimization, and application. Heat conduction from the molten weld pool to a base metal with a higher thermal conductivity absorbs thermal energy and determines the amount of energy needed to melt the base metal locally [9,10]. During cooling, the differing base metals’ thermal expansion coefficients produce tensile in one part and compressive stress in the other part. If the stress is not relieved, the metal deposited under tensile stress may experience hot cracking during welding or cold cracking during service [9,10]. When welding metals with significantly different thermal conductivity, the heat source must provide a thermal differential for proper thermal equilibrium. Heat conduction rapidly from the molten weld pool to the base metal with higher thermal conductivity takes away the thermal energy and affects the energy input required to melt the base metal locally. Thus, the heat source is always directed toward the metal with higher thermal conductivity to balance the heat. Preheating the base metal with a higher thermal conductivity can control heat loss to the base metal and reduce the cooling rate of the weld metal and the heat-affected zone (HAZ) [9,10]. During cooling, the difference in thermal expansion coefficients between the base metals creates tensile stress in one and compressive stress in the other. The metal deposited under tensile stress may experience hot cracking during welding or cold cracking during service if the stress is not released. This is critical in recirculation applications where the seal must operate at high temperatures.

### 2.2. Some Related Studies

Dissimilar welded joints between 92 and AISI 304L austenitic stainless steel were studied in an experimental investigation by Dak and Pandey [11]. The microstructure, mechanical properties, and residual stresses of the material were investigated in our study. The weldment’s microstructure and mechanical properties (tensile, Charpy impact, and microhardness) were determined using a scanning electron microscope and an optical, multi-pass gas tungsten arc welding (GTAW) method. A post-weld heat treatment (PWHT) at 760 °C for two hours, followed by air cooling, was used to homogenize the heterogeneous microstructure formed on the P92 side. As a result, this PWHT had no effect on the weld fusion zone or the SS304L heat-affected zone (HAZ) microstructure, but it did change the situation. Khan et al. [12], using shielded metal arc welding (SMAW) and tungsten inert gas (TIG) welding processes, investigated the effects of welding technique, filler metal, and post-weld heat treatment on stainless steel 304 and mild steel AISI 1020 dissimilar welding joints. Tensile and bending tests were performed to determine the best welding and PWHT method for this dissimilar junction. When comparing tensile and bending test results, it was discovered that a PWHT at 630 °C was the best heat treatment procedure for SMAW joints with both MS and SS electrodes. The optimal heat treatment process for TIG welding joints with both MS and SS filler materials was determined to be a PWHT at 600 °C. Ramakrishnan A. [13] experimented on the mechanical characteristics of TIG-welded dissimilar joints between AISI 304 and AISI 316 stainless steel using 308 filler rods. The effect of welding current on TIG welds has been studied before, and several mechanical tests have been carried out to validate the weld’s mechanical properties. We chose welding currents of 30, 45, and 60 amps. The chosen welding voltages were 40 V, 60 V, and 80 V. Weld quality was assessed using hardness and tensile tests. The weld zone had the highest hardness rating when compared to the heat-affected zone and base metal. When TIG-welded part with 60A of current, the specimen reached the highest ultimate tensile strength of 528.36 MPa. In an experimental study, Satputaley et al. [14] investigated the effect of TIG welding on the maximum tensile strength for 4130 Chromoly and 7075T6 aluminum. These results show that Chromoly 4130 has the same weld strength as Al 7075T6 (396.69 MPa), resulting in poor weld mechanical characteristics and welding performance. When utilized for high-strength applications, Al 7075T6 has a low weld penetration. TIG welding on Chromoly 4130 results in excellent weld penetration while maintaining the material’s properties. Ramadan and Boghdad [15] performed an experimental investigation on the parametric optimization of TIG welding influence with respect to the tensile strength of the dissimilar metals SS-304 and low-carbon steel by using the Taguchi approach and using process parameters such as current and gas flow rate. The weld quality was assessed using tensile strength. The tensile test results, which were welded by a parametric combination of variables, yielded the best value (8 gas flow rate and 120 current amperes). The effective welding parameters are current and gas flow rates were determined using the mean, signal-to-noise ratios, and the ANOVA test. An analysis and investigation on the weld characteristics of TIG welding with respect to the dissimilar metals SS304 and MS1040 was conducted by Vennimalai Rajan et al. [16]. Tensile strength, hardness, and bend tests inspected the weld quality. The results showed that the tensile strength was 507.902 N/mm^2^ at a peak load of 49.880 KN, and the bending strength was 31.213 N/mm^2^ at a peak load of 3.140 KN. The Rockwell hardness was 78 kgf. Anbarasu et al. [17], used mild steel (IS 2062) as a base material and super-duplex ER2594 as a filler material to investigate the effect of filler material on the hardness of TIG welds. Materials can be created by selecting current, welding speed, and gas flow as process parameters. The Taguchi method of the L9 orthogonal network was used to conduct the test. Microhardness and microstructure were two of the weld quality metrics. Using DOE Taguchi and ANOVA methodologies, Kausar [18] analyzed and studied the weld quality parameters of tungsten inert gas welding on different metal plates with SS316L and IS2062. The process parameters, such as current, voltage, and gas flow are chosen, and weld quality was determined by tensile strength, hardness, and microstructure. The findings demonstrated that voltage has the most significant impact on stiffness, followed by gas flow and current, and that the most important parameter is tensile current, followed by voltage and gas flow. The optimum parameters for hardness are 130 (amp), 60 (volts), and 9 (L/min) airflow, while the optimum parameters for tensile strength are 130 (amp), 50 (volts), and 10 (L/min) airflow. Good flow indicates that the connection is vital when it comes to microstructures. Hazari et al. [19] performed an experimental investigation of TIG welding on AA 6082 and AA 8011 by selecting the welding current, electrode diameter, and gas flow rate. The weld’s final quality was validated by its ultimate strength and hardness. The maximum tensile strength increased dramatically as the amperage was raised, but the yield strength improved significantly. By increasing the effectiveness of the shielding gas, the ultimate tensile strength increased according to the gas flow rate. The diameter of the filling material changed little; nevertheless, material soldered with a 2.5 mm diameter filler showed a significant difference. Using the TIG welding process, Sayed et al. [20] conducted a review paper of different metal welds in stainless steel and mild steel. A. Kumar et al. [21], used the Taguchi technique to conduct an experimental investigation of TIG welding on stainless steel 202 and stainless steel 410. Current, gas pressure and welding speed were the considered process characteristics, and the final tensile strength determined the quality of the weld. The primary impact graphs revealed that current, gas pressure, and welding speed were the significant influencing elements on the ultimate load. With a welding current of 130A, a gas pressure of 10 kg/cm^2^, and a welding speed of 2.8 mm/s, the ideal parameter setting for the ultimate tensile strength of 808.3 N/mm^2^ was determined experimentally. Devakumar [22] investigated GTAW welds on duplex stainless steel and hot-rolled steel (DSS/HRS) by characterizing the weld microstructure and examining the micro-composition by using energy-dispersive X-ray (EDAX) in order to determine metallurgical properties, and they used a microhardness test, tensile test, and bending test to determine mechanical properties. In contrast to the DSS parent metal, which has a grain austenitic with ferrite, and the HRS parent metal, which only has a long-grain similar to austenitic, the results showed the production of a triangular ferritic microstructure. The DSS/HRS weld was more efficient with fine ferrite grain. The creation of the dendritic delta ferrite microstructure appeared to be the result of the energy-dispersive X-ray analysis (EDAX). L. S. Kumar et al. [23] used the Taguchi technique for TIG and MIG welding to investigate the welding features of AISI 304 and 316. They discovered that, when welding austenitic stainless steels using the TIG method, the hardness value (BHN) at 40A for TIG is 162.3 and 196.54 for MIG. As a result, they concluded that the MIG technique is appropriate for low currents. They also concluded that the TIG welding pattern could sustain a final load of 57600N, whereas the MIG welding pattern could withstand 56160N. As a result, the TIG welding pattern can withstand more load than the MIG welding pattern. Thus, by observing the past work-performed on austenitic stainless steel. Some research focused on application, different grades of material, and welding parameter optimization. However, no attention was paid to SS316 and mild steel (dissimilar metals) in terms of the effect of filler metal type and root gap, as well as to parametric optimization during dissimilar metals welding for weld quality. Various researchers have experimented by considering common process parameters such as welding current, welding voltage, speed, and gas flow rate. Other process variables, such as filler wire and root gap, however, have a considerable impact on mechanical and microstructural qualities. Because the chemical and physical properties of materials differs when the base metals to be welded are dissimilar metals, the filler metals utilized compromise this difference. The gap between the roots is also a key determinant in weld penetration, which has an impact on mechanical properties. In the present work, parameters such as welding current, root gap, gas flow rate, and the type of filler metals were considered with respect to a similar and dissimilar TIG welding method using the Taguchi orthogonal array (OA) experimental plan. Finally, the influence of the parameters on weld quality was investigated with a signal-to-noise (S/N) ratio and analysis of variance (ANOVA), and then, the multi-objective parametric optimization was performed using Taguchi-based desirability function analysis (DFA) to find the optimum condition.

## 3. Materials and Method

### 3.1. Base Materials

The materials employed in this experiment were an austenitic chromium–nickel SS316 stainless steel and an AISI 1020 mild steel with a thickness of 3 mm, and a molybdenum concentration of 2% to 3% for the stainless steel. The inclusion of molybdenum improves the general corrosion resistance as well as strength at high temperatures. AISI 1020 is a low-carbon steel with a Brinell hardness range of 119 to 235 and a tensile strength range of 410 to 790 MPa [24]. Chemical composition of base and weld metal is shown in Table 1, and Figure 1 shows the spectrometry set-up.

### 3.2. Material Used for Filler

ER316, ER309 stainless steel, ER70S-3 mild steel, and ER312 stainless steel filler material with a diameter of 2 mm were employed as filler metals.

**ER-316L**: This filler metal has the same chemical composition as ER316, with the difference being that the carbon concentration is limited to 0.03 percent to prevent inter-granular carbide precipitation [25].

**ER-309L**: Is an austenitic stainless steel that can be used to weld 309, 309L, 304, 316, and other 300 series stainless steels.

**ER-312**: This stainless-steel electrode is the most versatile of the all-position electrodes. When the base metal is an unknown grade of steel, this electrode is ideal.

**ER70S-3**: Is adequate for general-purpose welding over clean to light levels of rust and mill scale [26]. Table 2 below shows the chemical composition of all filler metals. Figure 2a,b shows the electrode and filler metals used.

### 3.3. Experimental Set-up, Sample Preparation, and Welding Conditions

As illustrated in Figure 3, stainless sheets with dimensions of 100 mm × 75 mm × 3 mm were prepared using an E21S ESTUN shear cutting machine to produce stock sheet in sectioned plates. The edges of the work components were adequately prepared before welding. Using a wire brush, dust was removed from the edges and the space surrounding them. The process parameters and levels used in this study are shown in Table 3.

Experiments were conducted at the Centre of Excellence for Engineering (Welding Training Center) workshop on a TELWIN Superior 322 TIG welding machine using a direct current negative electrode, as per the Design of Experiments of Taguchi, and by considering process parameters like welding current, root gap, gas flow rate, and the type of filler metals at 4 levels, which derived from the L16 orthogonal array. Table 4 shows the L16 orthogonal array for the experiment. The welding setup have the following parts, as shown in Figure 4: TIG welding machine: During welding, current and voltage are delivered. Gas cylinder for purging gas: This makes use of argon gas, which is supplied into the welding flame at a precise flow rate to create an inert atmosphere and maintain a stable arc. Work holding table: A surface plate (made of cast iron) on which a work piece is secured so that the welding gap between the tungsten electrode and the work piece can be manually maintained between 1 to 3 mm. However, according to 16 combinations of data from the L16 OA, the welding current, root gap, gas flow rate, and type of filler metals vary for each test. The samples after welding are shown in Figure 5.

### 3.4. Weld Geometry and Defect Test

TIG weld quality is mostly determined by the geometry of the weld bead. Bead geometry is heavily influenced by welding process parameters, such as welding current, shielding gas flow rate, and gap distance. Variables such as the heat-affected zone, bead width, bead height, penetration, and area of penetration, as well as the mechanical qualities of the weld, such as tensile strength, hardness, and bending strength, are all crucial in defining the mechanical properties of the weld [27]. The bead width was determined by measuring the width of each welded sample with 0.01 mm least count vernier calipers. The weld with the least bead width was considered a good weld because the width of each welded sample varied based on the parameter used. Bead width of the weld is shown in Figure 6.

### 3.5. Liquid Penetrant Test (PT)

Liquid or dye penetrant testing (PT) is a nondestructive material-testing method that employs capillary forces to locate and see surface cracks or holes. It can identify faults on the surface, such as fractures, laps, and porosity. The penetration of liquid penetrant through open surface discontinuities is the basis of PT. Surface flaws, such as surface cracks, porosity, absence of fusion, and intergranular corrosion, are detected using the dye penetrant method. Welds and parent material were subjected to dye penetrant testing. PT testing can be used on any non-porous, clean material, whether metallic or nonmetallic; however, it is not recommended for extremely rough surfaces. When cleaning with chlorinated hydrocarbons, volatile petroleum distillates, or acetone, dwell time should be allowed for the evaporation of the volatile cleaner from discontinuities before the penetrant is applied. Then, the penetrant should be applied to the surface area of the test samples by spraying. After the penetrant was applied, some dwell time was allowed until the capillary action allowed it to permeate small fractures or pores. Finally, the test specimens were cleaned with solvents following completion of the examination. All processes and defects observed are shown below in Figure 7 and Figure 8, respectively.

### 3.6. Mechanical Property Testing

For various mechanical tests, TIG-welded experimental specimens were created from SS316 metals and machined using ASTM in order to determine the necessary dimensions for studying their properties.

### 3.7. Tensile Strength

Tensile testing is a fundamental materials science and engineering test in which a sample is subjected to a controlled tension until failure [28]. For conducting Tensile test HUT 600 servo-hydraulic with a piston speed of 8 mm/min and a max force of 600 KN; at the Ethiopian Technical Institute. Figure 9a–c shows the specimen prepared for tensile, welding setup, and the fractured specimens after tensile test, respectively.

### 3.8. Hardness Testing

In calibrated hardness-testing equipment, the hardness of a weldment or welded specimen is assessed by pushing a hardened steel ball or diamond point into a flat surface on the weldment or welded specimen with a prescribed load and then measuring the size of the resulting indentation [29]. The Vickers, Brinell, and Rockwell tests are the three most used hardness tests. The Rockwell hardness test is quick, inexpensive, and largely nondestructive, leaving just a tiny indentation on the specimen. ASTM E-18, as shown in Figure 10a,b, has three sample zones. The hardness strength of the welded samples was examined using BROOKS Rockwell hardness testing equipment with a ball shape indenter at a force of 100 kg for a duration of 10 s. The weld’s HRB hardness was assessed first with a standard Rockwell hardness tester with a ball shape indenter, and then the material’s hardness was determined with an EQUOTIP2 digital hardness tester that reads in HRB.

### 3.9. Flexural Testing

Flexural strength is a force (in newtons) per unit area that represents a material’s maximum stress tolerance [30]. A three-point flexure bending strength test was conducted with a steadily increasing load applied until the material broke or bent permanently. A flexural test machine may apply progressively higher levels of force while simultaneously recording the force at failure. Using a loading pin, the loading force was applied to the center. This arrangement ensured that the specimen is loaded evenly and that there is no friction between the specimen and the supporting pins [31]. The formula for calculating the flexural strength of a rectangular specimen in a 3-point test [29] is:σ = 3LF/(2bd^2^)(1)
where L = specimen length, F = total force applied to the specimen by two loading pins, b = specimen width, and d = specimen thickness. The sample rests on supports or on the bottom block and is not clamped during the test. This clearance is often the mandrel diameter, d, plus three times the specimen thickness, t. Bending force is applied to the center of the specimen [32]. The specimen under this test was sectioned with a power hacksaw that uses cooling fluid to keep the material properties cool (Center of Excellence for Engineering Workshop). A flexural test was performed at the Ethiopian Conformity Assessment Enterprise (ECAE) utilizing the servo-hydraulic universal testing machines with a cross head velocity of 10 mm/min at room temperature, as shown in Figure 11. The material test specimens for the 3-point bending test were manufactured in accordance with ASTM E290. The dimensions of each specimen was according to the standard. Figure 12a–c illustrates the welded specimens for the 3-point bending test, experimental work, and U-shape specimens after testing, respectively.

### 3.10. Microstructure and Microhardness Testing

Microstructure and microhardness in this study were computed at the Adama Science and Technology University (ASTU) using the HUVITZ HR-300 series optical microscopy model at the material science engineering laboratory. Scanning electron microscopy (SEM) (JCM-6000Plus scanning electron microscope) for microstructure, morphological, and fracture mechanisms was conducted on the weld zone at the biology laboratory. Microhardness testing was performed using a Vickers hardness testing machine at the material engineering laboratory. A Vickers hardness test was used for each specimen’s test. The resulting impression was measured and examined under a microscope before being turned into a hardness number [33]. For microstructural and microhardness testing, the samples were sectioned using an abrasive cutter with a coolant to protect the properties of the metal for testing. Then, the material was mounted. After mounting, the specimens were grinded using silicon carbide grit papers: #220-600-800-1200-2000. Furthermore, the samples were polished with a diamond suspension polishing agent using an electro-polishing machine. Finally, the specimens were etched for 15 s with an etchant solution containing 70% nitric acid (HNO_3_). Then, a metallographic test was performed. In the biology lab, scanning electron microscopy was performed to examine the microstructure using a JCM-6000Plus scanning electron microscope. Before sample preparation, the size, shape, condition, and conductivity properties of the sample were considered. The cross-section of the samples was 30 mm × 20 mm × 3 mm. The samples were thoroughly cleaned. Then, the samples were mounted and observed.

### 3.11. Taguchi Method

One of the design of experiments (DOE) application is Taguchi’s, which is used to examine the effect of a variety of parameters and their behavior by running a small number of experiments and analyzing their impact on a certain behavior. It can simultaneously select the ideal circumstances for the experiment, resulting in a superior result. There are three basic steps according to Taguchi: planning, conducting, and analysis [34]. Minitab 19 statistical software was used to develop the experimental design of the process parameters and numerical analysis. The total degrees of freedom provided must be more than the number of experiments required to study the factors [34]. In this study, a number of levels of various factors were used to identify which orthogonal array to use. Four factors were chosen for the experiments, each having four levels. Degree of freedom = P ∗ (L − 1), where P = number of factors, L = number of levels, and DOF = 4(4–1) = 12. Thus, the orthogonal array (OA) had 16 experimental runs. After calculating the degree of freedom (DOF), the number of experimental runs was chosen from an orthogonal array. The Taguchi technique highlights the importance of using the signal-to-noise ratio to evaluate response variation, which decreases the impact of uncontrollable parameter variation on quality features. The desired value (mean) for the output characteristic is referred to as signal, while the undesired value (deviation, SD) is referred to as noise in the Taguchi method. As a result, the S/N ratio equals the SD ratio. Taguchi uses the S/N ratio to assess the quality of attributes that deviate from the desired value. Depending on the type of characteristic, many S/N ratios are available; smaller is better, nominal is best, and larger is best [35,36]. Optimization of the multi-response problem is a challenge with respect to optimizing output responses all together. Among the various simultaneous optimization methods, most of the authors used approaches that combine all the different response requirements into one composite requirement. Hence, a compromise solution is obtained in a much simpler way [37]. Harrington invented the desirability function to simultaneously optimize many responses, and desirability function analysis (DFA) is a widely used technique in academics and industry for the simultaneous optimization of several responses [38]. The Derringer and Suich methods were eventually modified to increase their usefulness, and they became quite popular [39]. The desirability scale ranges from 0 to 1, and it measures how near an answer is to its ideal value. The desirability of a response is 0 if it falls within the unacceptable intervals, and 1 if it falls within the ideal intervals or the response reaches its ideal value. The closer the response comes to the ideal intervals or ideal values, the more desirable it becomes. According to the objective features of a desirability function, the nominal-the-best (NB) answer, the larger-the-better (LB) response, and the smaller-the-better (SB) response can be classified. A composite desirability function can be created by combining all the desirability functions to turn a multi-response problem into a single-response problem.

### 3.12. ANOVA (Analysis of Variance)

Analysis of variance and analysis of a response parameter for numerous experimental responses were calculated using the Minitab19 statistical software. The influence of all control values was determined using the statistical approach of ANOVA. The percentage contributions of each control factor were utilized to measure the associated impacts on the performance characteristics in the analysis of variance. The significance level was set at 5%, which corresponded to a 95% level of confidence. The following formula was used to obtain the percentage contribution of each influential parameter from the analysis of variance [34].
(2)$$P\left(\%\right)=\frac{\mathrm{SSTR}}{\mathrm{SST}}*100$$
where P = percentage contribution of individual factors, SSTR = sum of squares, and SST = total sum of squares [34].

## 4. Results and Discussion

During the TIG welding process, the influence of welding parameters on mechanical properties, welding joint shape, and the microstructural properties of the dissimilar weld of SS316 stainless steel and AISI 1020 mild steels was investigated. After the welding process was completed, the welded samples were visually inspected for welding defects using liquid penetrant.

### 4.1. Liquid Penetrant and Visual Defect Testing

Table 5 below shows the PT test results for the specimens. While the welding was being performed, all the samples were visually inspected. A few welding faults, such as a bit of undercut at the end, excessive reinforcement height, lack of fusion, porosity, root crack, and lack of penetration, were found in some of the welded samples. These samples—numbers 1, 2, 5, 9, and 13—had problems, but the rest of the samples are defect-free.

**Lack of fusion:** There was a lack of fusion in sample numbers 2 and 5. This defect occurred due to minimum current input, arc length, electrode angle, electrode manipulation, and improper welding parameter settings, which contributed to the fusion failure. Lack of fusion can also be caused by improperly cleaning oily or scaled surfaces. These defects are more prone to appear during functionality [29]. **Porosity:** This defect was observed in sample number 13. This defect was created by air entrapped in the shielding gas, which resulted in dispersed porosity and gross surface pore breaking. Porosity can be caused by leaks in the gas line, too high a gas flow rate, draughts, and excessive turbulence in the weld pool. **Undercut:** An undercut defect was observed in sample number 1. At the weld, undercut discontinuities formed a mechanical notch interface [29].

### 4.2. Microstructural Analysis of Welded Metals

An optical microscope (model: HUVITZ HR-300 series) was used to examine the microstructure of all the TIG-welded samples, as well as the base material. However, only two specimens were supplied with base materials: sample number 10 and sample number 1. These samples were chosen for microstructural investigation based on their tensile test results. Sample number 10 indicated a maximum UTS joint strength weld, whereas sample number 1 showed a minimum UTS joint strength.

Figure 13 shows a typical optical micrograph of the base metal. A totally austenitic structure with small annealing twins can be found in the microstructure of 316 stainless steels, as shown in Figure 13a. Similar results have also been reported for stainless steel 316 [40]. The structure of the low-carbon steel, AISI 1020, on the other hand, showed a mixture of pearlite and a dominant ferritic phase (Figure 13b). Figure 14 and Figure 15 show the optical microstructures of TIG-welded joints and the weld metal from the selected samples. In all sample micrograph pictures, the base metal (BM), heat-affected zone (HAZ), and weld metal (WM) can be easily differentiated. Sample number 10 was welded with AWS-ER309L filler wires, resulting in a microstructure of ferrite and austenite (i.e., a ferrite structure), which could be explained by the principal solidification modes of the weld metals. The Creq/Nieq ratio explains the weld metal solidification mode. The Schaeffler formula was used to calculate the Creq/Nieq ratio [29]:(3)Creq/Nieq ratio=Creq of weld metalNieq of weld metalSample 10-Creq/Nieq ratio=17.999.81=1.83Sample 1-Creq/Nieq ratio=2.841.89=1.50
Therefore, the solidification mode is ferritic–austenitic for all samples; that is given by:FA = 1.48 ≤ *Creq*/*Nieq* ≤ 1.95(4)

The chemical composition of both the weld metals and the type of filler wires used is listed in Table 2.

The weld zone of the welded specimens in sample number 10 exhibits a very fine delta ferrite along the sub-grain and migrating boundaries in the plain austenitic matrix due to moderate heat input and a moderate cooling rate, as shown in Figure 14; ref. [41] reported the same result. In this weld zone, the delta ferrite (δ) is finer; the heat provided by the welding process, which was modest (110A), is related to the delta ferrite’s (δ) size. When heat input is modulated, a modulated cooling rate occurs, and a finer skeletal ferrite is generated as a result. Due to very low welding heat inputs, i.e., high cooling rates (70A), the dissimilar weld metals in sample number 1 included δ-ferrite in the form of dendritic lathy-ferrite at the dendrite core surrounded by an inter-dendritic γ-phase.

### 4.3. Fractography Analysis

Scanning electron microscopy (SEM) was used to take micrographs of the break zone in the tensile test. According to our findings, the fracture location of the welds was in the base metal of all the welded specimens. The specimens chosen for the micrograph testing of fracture mechanisms were those with the highest ultimate tensile test results. The sample’s weld penetration was fully penetrated, as shown in Figure 16, and the weld zone was found to have a good interface with the base metals. The dissimilar weld fracture micrographs revealed the same profile with the production of dimple, indicating that the fracture was in the ductile mode, as illustrated in Figure 17. It was discovered that the width of the weld-generated had the quality of full penetration from the face pass to the root pass. Weld bead shape can be better controlled via moderate current input utilizing filler metals, such as ER-316. As seen in [42], as the input was very low, the weld pool depth and width were very small (low penetration); however, at a moderate current, the weld pool width and welded area were controlled. AWS ER-309L filler is recommended for the dissimilar welding of stainless steel to low-carbon steels [32].

### 4.4. Microhardness Analysis

Figure 18 depicts the differences in Vickers microhardness values as a function of the distance from the weld center to the base metals on both sides of sample 14. The hardness values of all the weld areas were clearly higher than the hardness of the base metals. Welded area has the highest hardness. The average hardness value in the fusion zone of the welded metals was 183 HV or 90 HRB. In comparison to the hardness values of the base metals, the HAZ sides had higher hardness values. The hardness values in the HAZs of the 316 stainless steel welds were 162 HV or 85 HRB, and HAZ of the AISI 1020 base metals was 147 HV or 77 HRB.

### 4.5. Mechanical Properties

The purpose of this research was to investigate and optimize TIG process parameters for dissimilar welding between AISI 316 and AISI 1020. We did so by analyzing the microstructure, mechanical properties (hardness, tensile, flexural bending, properties), and bead width of weldments produced with these techniques. Following that, the test response parameters were analyzed using the Taguchi method, and the percentage contribution of each parameter was determined using analysis of variance (ANOVA). Finally, the parameters were optimized with a multi-response optimization technique using a desirability function analysis (DFA). The welding control parameters and levels utilized are shown in Table 6, below.

### 4.6. Tensile Strength Result Analysis

Tensile strength data were collected using a universal testing machine. Each test consisted of two specimens, with the average value taken. For each trial experiment, Table 7 presents the measured experimental results of the mean tensile strength values. The highest affected parameters, as shown in Table 8, were welding current (A) at level three, followed by gas flow rate (B) at level three, filler type (D) at level three, and root gap (C) at level three. Taguchi used these welding parameters (A3B3C4D3) as the initial parametric setup for the TIG-welding of stainless steel 316 with a thickness of 3 mm. This implies that it is possible to employ that as a parameter combination in order to generate larger and better outcomes.

In the main effect plot graph below, the numerical number at the maximum point in each graph represents the parameter’s best value at that level. They also display the optimum settings within the experiment’s parameters. When compared to other welding variables, the welding current weight percent, followed by the gas flow rate, had the largest impact on improving tensile strength. The ideal parameters (A3B3C4D3) were welding current (110A) at level four, gas flow rate (13 L/min) at level three, root gap (2.5 mm) at level four, and filler type (ER-309) at level three. Taguchi used the S/N ratio to evaluate quality attributes that differed from the desired value. As a result, the larger the tensile strength, the better the quality. Based on the achieved tensile strength result, the effect of the above parameters on tensile strength was examined.

According to the signal-to-noise ratio value in Table 9, the welding current weight percentage has the greatest influence, (Delta = 1.17; Rank = 1st) followed by gas flow rate (Delta = 0.92; Rank = 2nd) and filler metal (Delta = 0.85; Rank = 3rd). The (S/N) ratio is least affected by the root gap (Delta = 0.52; Rank = 4th) according to the experimental result. The initial best parameter combination for increasing weld metal stiffness with the greater-the-better responsiveness of TIG-welded 316 stainless steel sheets is **A3B3C4D3**.

The effect of these factors on tensile strength is shown in Figure 19. The tensile strength increases dramatically as the current climbs from level one to level three, as shown in the graph above. The tensile strength decreases drastically as the current is increased from level three to level four. As the gas flow rate increases from level one to level three, the tensile strength value increases; ref. [43] reported the same result. However, as the gas flow is increased from level two to level four, the material’s tensile characteristics decrease. The root gap was another important factor; when the root gap expanded from level one to two, the tensile strength increased from a low to a high number. The tensile characteristic of the metal decreases as gaps increase from level two to level three. However, widening the gap from level three to level four maximized the material’s tensile property [44]. The third most important welding attribute was the filler metal type; level one and two filler metal had nearly identical tensile strengths. The tensile strength of level three filler was the highest of all the fillers employed. However, the filler at level four was worthless.

The adequacy of the developed models was tested using ANOVA. As per this technique, if the calculated value of the F-ratio of the developed model is less than the standard F-ratio (F-table value 3.49) value at the desired level of confidence of 95%, then the model is said to be adequate within the confidence limit. The coefficient of determination, “R2”, for the ANOVA found that it was above 0.99. The most significant parameters that affect the tensile strength of TIG-welded steels are identified using analysis of variance (ANOVA), as shown in Table 10.

Since the F-value from the table is 3.49, but the F-ratio gained from the ANOVA table is much greater than the critical F-ratio, this means that all parameters have a significant effect on tensile strength. As shown in the table above, the welding current weight percentage (40.8%), gas flow rate (29%), filler metal (19.80), and root gap (9.90) all have a substantial impact on the tensile strength of TIG-welded 316 stainless steel sheets with a thickness of 3 mm.

### 4.7. Hardness Test Results Analysis

Hardness measurement data were obtained using a Rockwell B hardness testing machine, with each test performed on the base metal HAZ and the weld zone and each at 3 mm from the weld zone’s center. In each zone, three readings were recorded, and the average value was calculated. Table 11 shows the measured experimental results of the mean hardness values for each trial experiment. Table 12 reveals that welding current (A) at level 4, root gap (C) at level 3, gas flow rate at level 4, and filler type at level 4 have the maximum effect on hardness. These welding parameters (A4B4C3D4) were chosen as the first parametric setup for hardness when TIG-welding stainless steel 316 to AISI 1020 mild steel with a thickness of 3 mm according to the response data means of the experiment. As seen from the signal-to-noise ratio value in Table 13, the welding current weight percentage had the greatest impact (Delta = 1.69; Rank = 1st) followed by root gap (Delta = 0.72; Rank = 2nd), and gas flow rate (Delta = 0.61; Rank = 3rd). The filler type had the smallest impact on the S/N ratio (Delta = 0.42; Rank = 4th). The Taguchi initial optimum parameter combination for boosting the hardness of dissimilar TIG-welded 316 stainless steel and AISI 1020 mild steel is **A4B4C3D4** based on the larger-the-better characteristic.

In the parameter effect plot graph of the S/N ratio, the numerical number at the minimum point in each graph represents the parameter’s best value at that level. They also display the optimum settings within the experiment’s parameters. When compared to other welding variables, the welding current weight percent, followed by the root gap, has the biggest impact on improving hardness. The influence of welding settings on hardness is depicted in Figure 20, as well as the optimal values depending on their rank. The optimal values for maximizing hardness are current at level 4 (130A), gas flow rate at level 1 (16 L/min), root gap at level three (2 mm), and filler metal at level two (SS 312) according to the analysis and the primary effect plot graph for the S/N ratios. As the current rises from level one to level four, the hardness rises proportionally with it, as indicated in the graph above. The harder the metal was at level four as shown in [23]. The material’s hardness increases as the gas flow rate increases from level one to level four. The same result was identified by [17] for mild steel using duplex stainless-steel filler. The second influential parameter was the root gap; as the root gap climbed from one to three, the hardness increased from the lower limit to the maximum value. When the gap reached level four, the hardness of the metal began to decrease. However, according to the experimental results, the gap at level three provided the optimal opening between the base metals. The fourth and most essential welding attribute was the filler metal type; when compared to other filler metals, level four filler metal had a greater hardness value, followed by the level three and level two fillers. The listed hardness value is in the filler at level one.

Using analysis of variance (ANOVA), the most important characteristics (most significant parameters) that determine the hardness of TIG-welded steels are found and displayed in Table 14. To determine the percentage contribution of each of the components, a study was conducted using their S/N ratios.

Since the F-ratio of the ANOVA table shown above was higher than the F-critical, all the parameters had a highly significant effect on hardness. The welding current weight percentage (74.57%), gas flow rate (8.52%), filler metal (4.50), and root gap (12.52) had a considerable impact on the hardness of TIG-welded 316 stainless steel and AISI 1020 mild steel sheets with a thickness of 3 mm.

### 4.8. Flexural Strength Result Analysis

The maximum breaking force was recorded from a three-point bend test under the universal testing machine, and the flexural strength was computed using this force and cross-section area. Three readings were taken for each testing trial, and the average value was calculated. Table 15 shows the measured experimental results of the mean flexural strength values for each trial experiment.

Table 16 shows that the welding current (A) at level two has the highest value, followed by filler metal (D) at level three, root gap (C) at level four, and gas flow rate (B) at level two. For the TIG-welding of stainless steel 316 to AISI 1020 mild steel with a thickness of 3 mm, these welding parameters (A2B2C4D3) were chosen as the initial parametric setup. This suggests that it is possible to use that as a parameter combination to generate larger, better outcomes.

The most important factor is the welding current weight % (Delta = 1.88; Rank = 1st) followed by filler metal type (Delta = 1.81; Rank = 2nd), and root gap (Delta = 1.21; Rank = 3rd) as shown in Table 17. The less influential effect on the S/N ratio is the gas flow rate (Delta = 0.68; Rank = 4th). Based on the larger-the-better flexural strength characteristic, the initial optimum parameter combination for boosting the rupture strength of TIG-welded 316 stainless steel with AISI 1020 mild sheets of steel was **A2B2C4D3**.

Figure 21 depicts the effects of welding factors on flexural strength, as well as the optimal values for each rank. According to the analysis and the primary effect plot graph for the S/N ratios, the optimal values for maximizing flexural strength are current at level two (90A), gas flow rate at level two (10 L/min), root gap at level four (2.5 mm), and filler metal at level three (SS 309), with an initial parametric setting of A2B1C3D3.

The numerical value at the greatest point in each graph represents the parameter’s best value at that level, as seen in the main effect plot graph below. They also show the best options within the parameters of the trial. When compared to other welding variables, the welding current weight percent had the greatest impact on improving flexural strength, while the gas flow rate had the least. Welding current (90A) at level two, gas flow rate (10 L/min) at level two, filler type (ER-309) at level three, and root gap (2.5 mm) at level four (which had a combination of factors) were the best (A2B2C4D3). As the current rises from level one to level two, as illustrated in the graph above, the flexural strength increases proportionally with the current input. As the current is increased from level two to level three, the flexural strength of the metal backs decreases significantly, while the rupture strength of the metal backs increases; ref. [16] reported the same value. The flexural strength of the material increases as the gas flow rate increases from level one to level two. As you progress through the levels, the rupture strength decreases. When the purging gas was increased to level four, however, the material strength began to increase again. The root gap was the most important factor; as the root gap went from one to two, the flexural strength decreased significantly. When the gap between levels two and four was increased, the flexural strength increased dramatically. The filler metal type was the second most important welding attribute; level one filler metal had a lower rupture strength rating than other filler metals. The level three filler had the maximum rupture strength, followed by the level two and four fillers.

ANOVA was used to assess the suitability of the produced models, as shown in Table 18. Since the determined value of the F-ratio of the produced model was less than the standard F-ratio (F-table value 3.49) value at the desired level of confidence of 95%, the model is inadequate within the confidence limit according to this technique. The coefficient of determination, or R^2^, is greater than 0.99.

According to the experimental results, all parameters had a significant effect on flexural strength because F-critical = 3.49 is less than the F-ratio. The contribution of each parameter, the welding current (46.57 percent), filler metal type (32.58 percent), and root gap (15.8%) have a highly significant impact, while the gas flow rate (4.80 percent) has a significant impact on the flexural strength of TIG-welded 316 stainless steel and AISI 1020 mild steel sheets.

### 4.9. Bead Width Result Analysis

The welded metal’s bead width was determined by measuring the width of the material deposited between the base metals with a vernier caliper with a 0.01 precision. In each welded specimen, three readings were collected, and the average value was calculated. Table 19 shows the measured experimental results of the mean bead width geometry values. Table 20 shows current (A) at level two, followed by root gap (C) at level three, filler type (D) at level two, and gas flow rate (B) at level four. These welding settings (A2B4C3D2) were chosen as the initial parametric setup from the results of TIG-welding of stainless steel 316 to AISI 1020 steel with a thickness of 3 mm. This implies that using this as a parameter combination to create smaller but better results is viable.

In the main effect plot graph above, the numerical number at the minimum point in each graph represents the parameter’s best value at that level. They also display the optimal settings within the experiment’s parameters. When compared to other welding variables, the welding current weight percent, followed by the root gap, has the biggest impact on enhancing bead width. The best settings were welding current (90A) at level one, gas flow rate (16 L/min) at level four, root gap (2 mm) at level three, and filler type (ER-316) at level two. According to the signal-to-noise ratio value shown in Table 21, the welding current weight percentage had the most impact (Delta = 1.01; Rank = 1st) followed by root gap (Delta = 0.81; Rank = 2nd), and gas flow rate (Delta = 0.61; Rank = 3rd). The filler type had the smallest impact on the S/N ratio (Delta = 0.49; Rank = 4th), as shown in Table 21.

Figure 22 demonstrates how welding factors influence bead width, as well as the optimal values for each factor depending on their rank. The optimal values for minimizing bead width are current at level two (90A), gas flow rate at level four (16 L/min), root gap at level three (2 mm), and filler metal at level two (SS 316), according to the analysis and primary effect plot graph for signal-to-noise ratios. The bead diameter of the weld decreases as the current climbs from level one to level two, as shown in the graph above. The bead width (weld width) increases dramatically as the current is increased from level two to level four, reaching its maximum value; ref. [45] reported a similar identification. As the gas flow rate rises from level one to level two, the width of the weld widens, which is undesirable. The bead of the weld narrows substantially as the gas flow rate climbs from level two to level four, and the gas flow rate at level four was the optimal flow for minimal bead width. The root gap was the second most important process parameter; when the root gap increased from level one to three, the bead width fell from a larger to a smaller number. The bead width grows gradually as gaps rise from level three to level four. As a result, at stage three, the ideal bead width was chosen. The list-relevant welding property was the filler metal type; the level-two filler metal had a narrower bead width value than other filler metals. The level-four filler metal had the largest bead width value, followed by the level-three and -one filler metals.

The determined F-ratio of the produced model was less than the standard F-ratio (F-table value 3.49) value at the required degree of confidence of 95 percent; thus, the model was deemed to be inadequate within the confidence limit according to this technique. The coefficient of determination, or R^2^, was determined to be greater than 0.98, as shown in Table 22.

The F-value = 3.49, but the F-ratio from the ANOVA table is much higher than the critical F-value; thus, the parameters used under this investigation have a high significant on weld bead.

### 4.10. Desirability Function Analysis (DFA)

In this study, Taguchi methodology was used to determine the best process parameters for single performance characteristics only, whereas desirability function analysis was used to determine the best process parameters for multiple performance characteristics at the same time. The Single response parametric setup was computed using the Taguchi technique to find the multi-response parametric setup with desirability; thus, these values were used in each result and in the analysis of tensile strength, hardness, flexural strength, and bead width. There are six steps in desirability function analysis.

**First step:** Formatting the response parameter result. There are four responses in Table 23; however, the expected positive reaction in each response was different depending on the aims that were needed.

**Second and third step:** The individual and composite desirability index are computed, respectively, as illustrated in Table 24.

**Fourth step:** The optimal parameter and its level combination were determined, as shown in Table 25. The higher composite desirability value implied better product quality. Taguchi’s experimental work was conducted with this CDI value as the response parameter to obtain the best parametric configuration. The Taguchi method was then used to analyze the composite desirability values. The means of the di and dg values are shown in Table 26. On multiple responses, the main effect plot was used to find the main influencing (most significant) parameter. Figure 23 shows that the highest influencing parameter for dg was root gap, followed by welding current and filler type.

**Fifth step:** The analysis of variance (ANOVA) was computed for CDI (dg). Using analysis of variance (ANOVA), the most important characteristics that determine the composite desirability index of TIG-welded dissimilar steels are found. To determine the percentage contribution of each component, a study was conducted using their S/N ratios. According to the CDI ANOVA table, root spacing and welding current had the greatest contribution, followed by filler wire type, as shown in Table 27.

The root gap (48%) and welding current (35) had a substantial influence, and filler metal (15%) had a significant impact on the composite desirability CDI (dg) of TIG-welded dissimilarly joined 316 stainless-steel sheets and AISI 1020 mild steels with a thickness of 3 mm.

**Sixth step:** Finally, we predicted and validated the quality attributes using the ideal level of design parameters, as indicated in Table 28.

The improvement of the parametric optimization between the initial and optimal setting is shown in Table 29.

### 4.11. Confirmation Test

A confirmation test was required after the experimental optimization to validate the optimized parametric configuration. As a result, the confirmation was carried out at the Ethiopian Technical Institute’s Engineering Center of Excellence at the welding training center. The results of the confirmation testing agreed with the predicted optimum parametric setting result with a very minimal positive validation error, as shown in Table 30.

## 5. Conclusions

Using welding factors such as welding current, gas flow rate, root gap, and filler type, this study studied and optimized the parameters of tungsten inert gas welding with respect to dissimilar (SS316 and AISI 1020) steels. A Taguchi L16 orthogonal array and the signal-to-noise ratio method were utilized to analyze the experimental findings, and ANOVA was employed to assess the influence of each parameter. DFA was used to undertake multi-response optimizations of the input parameters to reduce their impact on the mechanical and microstructural characteristics, as well as bead shape. The following conclusions were formed as a result of the experimental findings:

For welding flaws, nondestructive testing was used, and problems such as porosity, lack of fusion, and lack of penetration were discovered in a few samples. Due to a moderate heat input and moderate cooling rate, the weld zone of sample number 10 welded with ER-309L filler wire, which revealed a very fine skeletal ferrite along the grain, sub-grain, and migrating boundaries in the plain austenitic matrix. Sample number 1, which were welded using ER-70S-3 filler wire, had δ-ferrite in the form of dendritic lathy δ-ferrite in the dendrite core surrounded by an inter-dendritic γ-phase due to very low welding heat inputs, i.e., high cooling rates. The parametric combination of 110A welding current, 10 L/min gas flow rate, 2.5 mm root gap, and ER-309L filler wire resulted in a higher tensile strength of 502. The most significant characteristic for tensile strength was welding current, followed by gas flow rate, according to ANOVA. A maximum hardness of 90HRB or 183HV was achieved using a parametric setup of 130A welding current, 10 L/min gas flow rate, 1.5 mm root gap, and ER-312 filler wire. According to an analysis of variance (ANOVA), the most critical characteristic for hardness was welding current, followed by root gap. A maximum flexural strength of 692 was achieved using a parametric setup of 90 welding current, 7 L/min gas flow rate, 1.5 mm root gap, and ER-309L filler metals. According to an analysis of variance (ANOVA), welding current was the most important factor in flexural strength, followed by filler type. A parametric combination of 90A welding current, 16 L/min gas flow rate, 2 mm root gap, and ER-316L filler metals was used to obtain a minimum weld width of 6.21 mm. The most important characteristic on weld width was welding current, followed by root gap, according to an analysis of variance (ANOVA). According to the optimization results, all responses improved from the initial value to the optimum parametric setting, which increased UTS from 445 MPa to 493.2 MPa (10.83%), hardness from 83 HRB to 86 HRB (3.61%), flexural strength from 679 MPa to 691 MPa (1.77%), bead width from 6.21 mm to 5.8 mm (6.60%), and overall composite desirability from 0.7638 to 0.9387 (22. (**A2B2C4D3**)). This parameter combination matched the results of the confirmation test.

## Figures and Tables

**Figure 1 materials-15-04426-f001:**
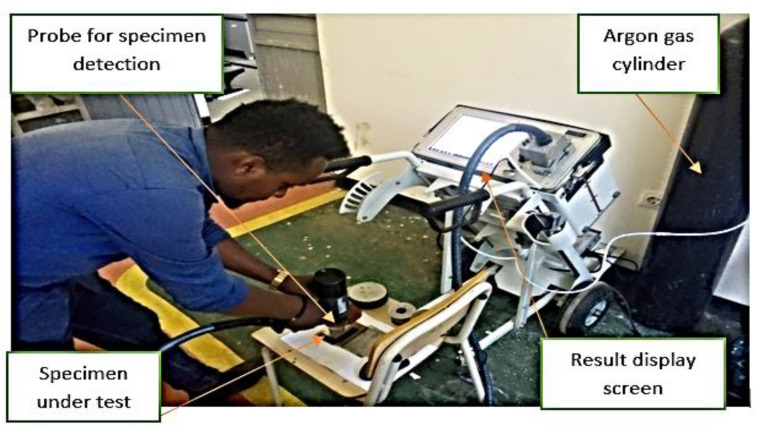
Setup ARC-MET8000 spectrometer (Ethiopian Technical University).

**Figure 2 materials-15-04426-f002:**
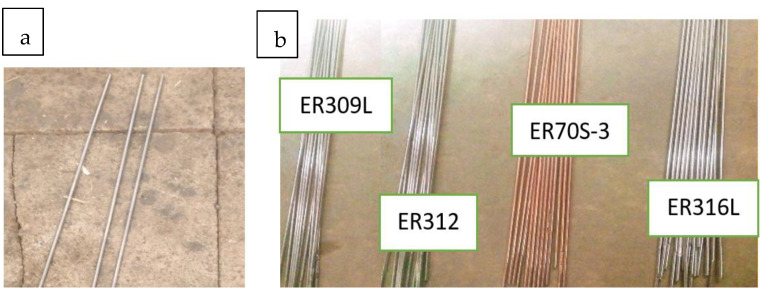
Material used: (**a**) tungsten (98%) with (2%) thorium oxide electrode; (**b**) filler metals.

**Figure 3 materials-15-04426-f003:**
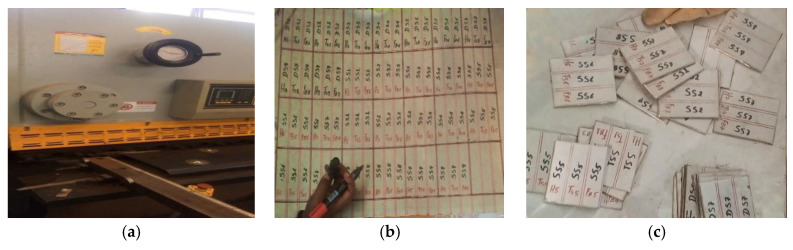
Specimen preparation: (**a**) shear cutting machine, (**b**) stainless steel, (**c**) after shear cutting.

**Figure 4 materials-15-04426-f004:**
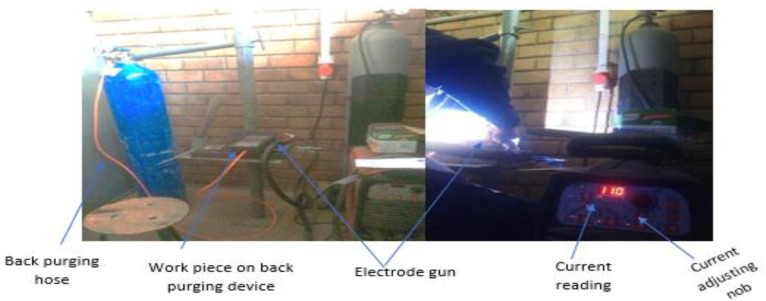
GTAW welding setup (Centre of Excellence for Engineering, Addis Ababa, Ethiopia).

**Figure 5 materials-15-04426-f005:**
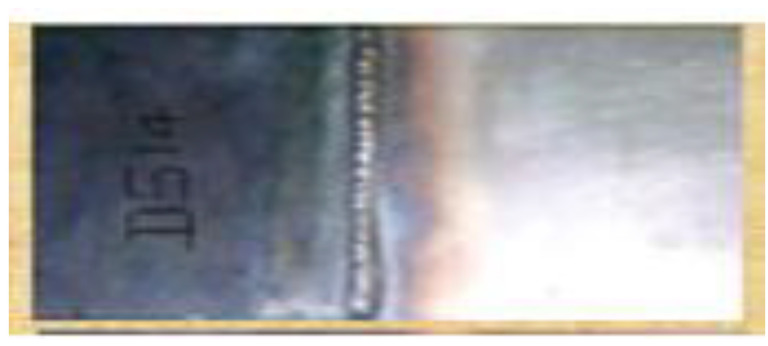
Samples after welding.

**Figure 6 materials-15-04426-f006:**
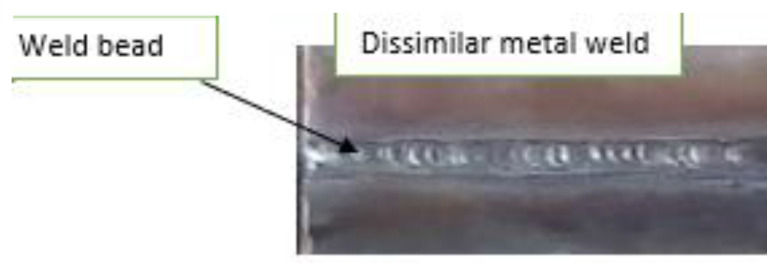
Weld bead.

**Figure 7 materials-15-04426-f007:**
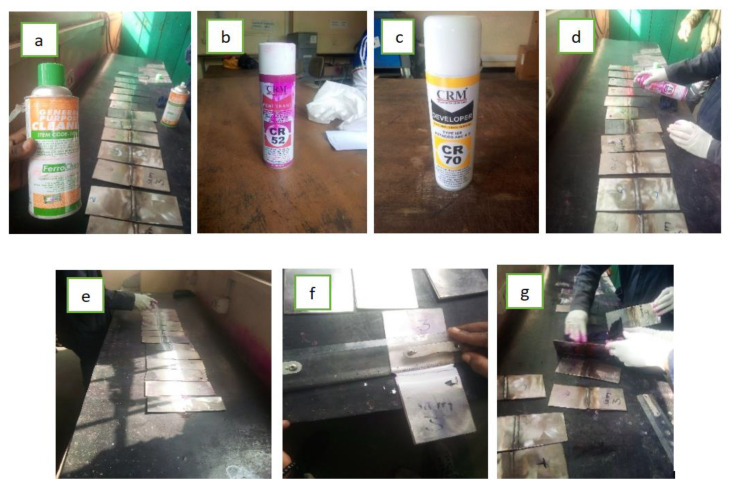
PT processes: (**a**) precleaning samples by using general purpose cleaner; (**b**) penetrant for segregation; (**c**) developer; (**d**) application of a penetrant liquid; (**e**) application of developer; (**f**) inspection of test surface; (**g**) post-inspection cleaning (anticorrosion solutions).

**Figure 8 materials-15-04426-f008:**
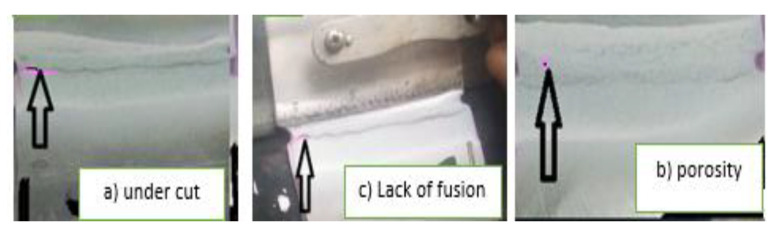
Defects in dissimilar weld metals: (**a**) sample no. 1, (**b**) sample no. 2, and (**c**) sample no. 13.

**Figure 9 materials-15-04426-f009:**
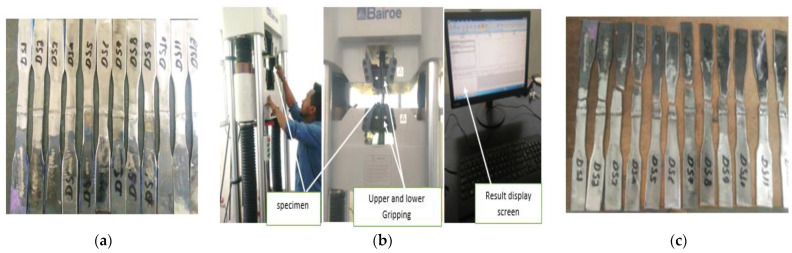
Specimen prepared for tensile strength, (**a**)Tensile Specimens (**b**) set-up, and (**c**) fractured specimens.

**Figure 10 materials-15-04426-f010:**
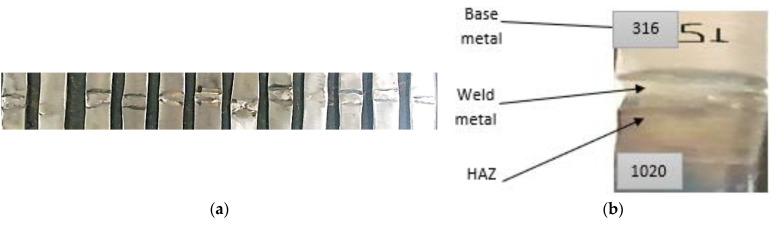
(**a**) Specimen prepared for hardness test; (**b**) three zones of samples.

**Figure 11 materials-15-04426-f011:**
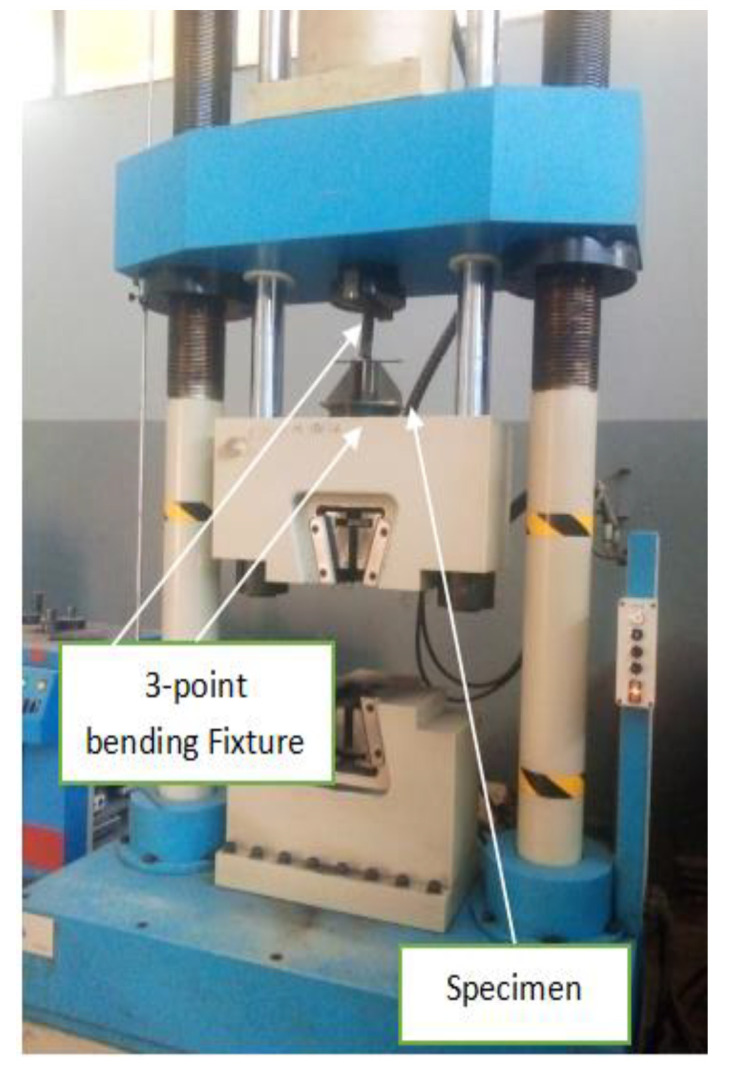
Flexural testing setup (Ethiopian Conformity Assessment Enterprise).

**Figure 12 materials-15-04426-f012:**
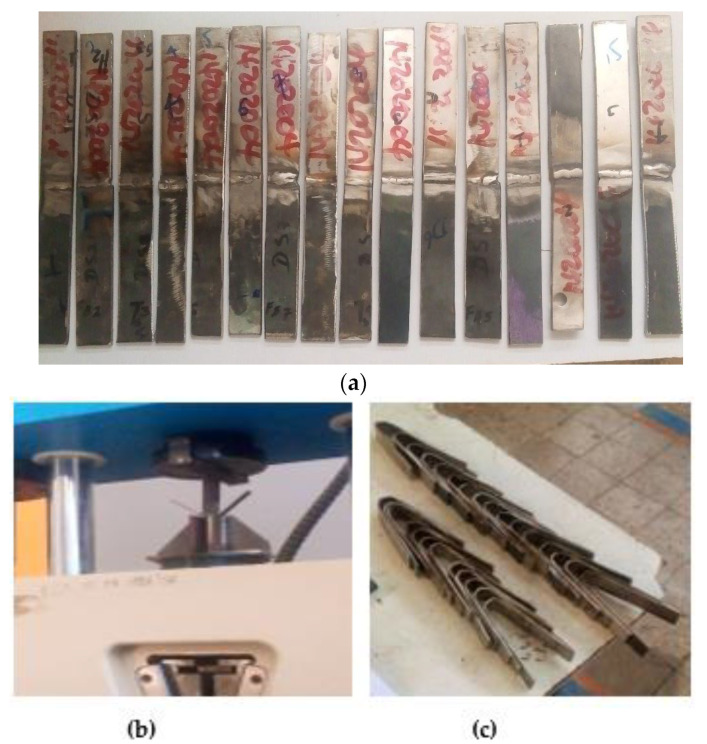
Specimens: (**a**) before testing, (**b**) specimen under 3-point flexural testing, and (**c**) specimens after testing.

**Figure 13 materials-15-04426-f013:**
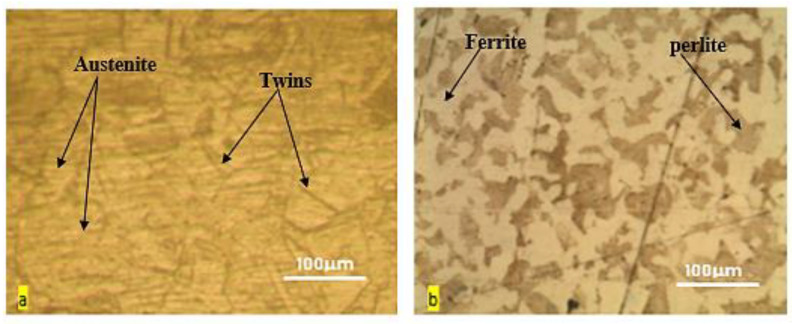
Microstructure of base metals: (**a**) 316 stainless steels; (**b**) AISI 1020.

**Figure 14 materials-15-04426-f014:**
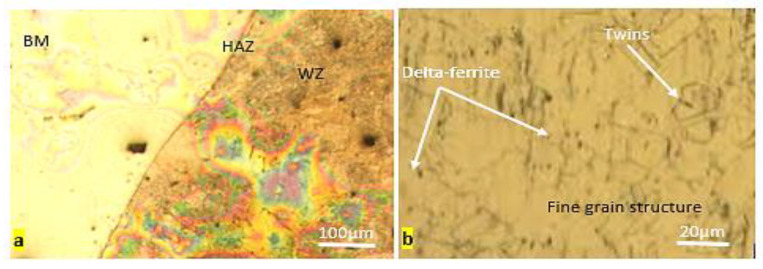
Microstructure of welded joint made of SS316 and AISI 1020 (sample 10): (**a**) welded joint; (**b**) weld metal.

**Figure 15 materials-15-04426-f015:**
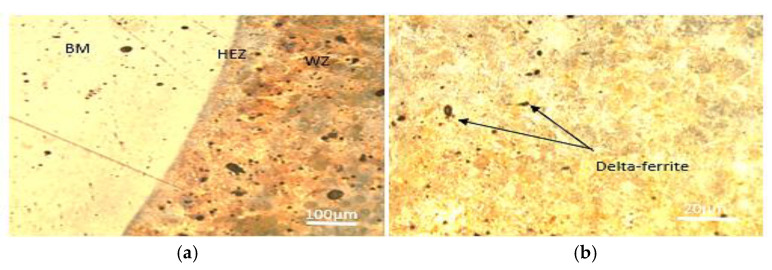
Microstructure of welded joint made of SS316 (sample 1): (**a**) welded joint; (**b**) weld metal.

**Figure 16 materials-15-04426-f016:**
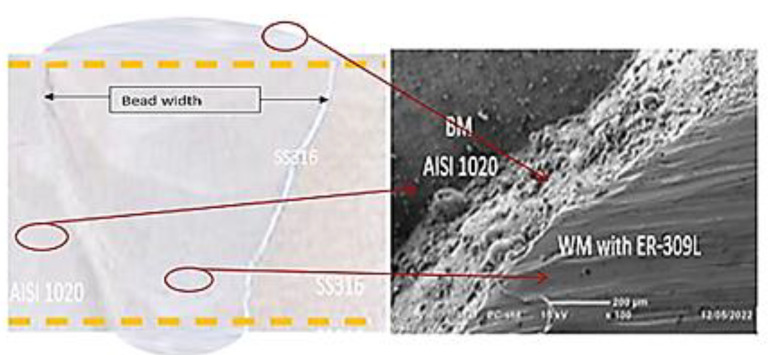
Weld bead of welded metals specimens welded image with ER-309L.

**Figure 17 materials-15-04426-f017:**
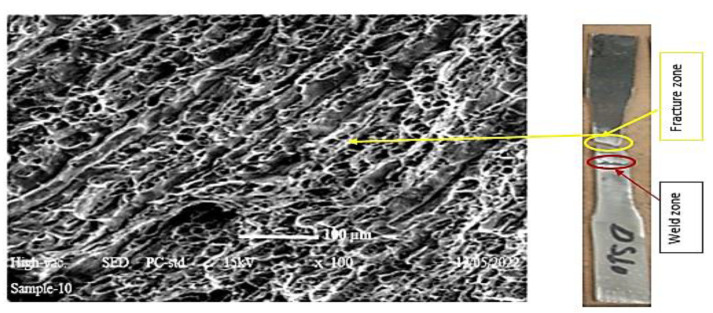
Tensile fracture surface obtained in the base metal of 316 stainless steel.

**Figure 18 materials-15-04426-f018:**
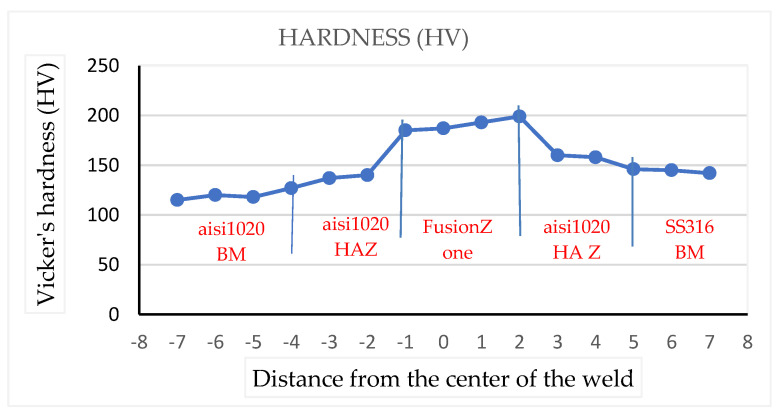
Vickers hardness of sample no. 14.

**Figure 19 materials-15-04426-f019:**
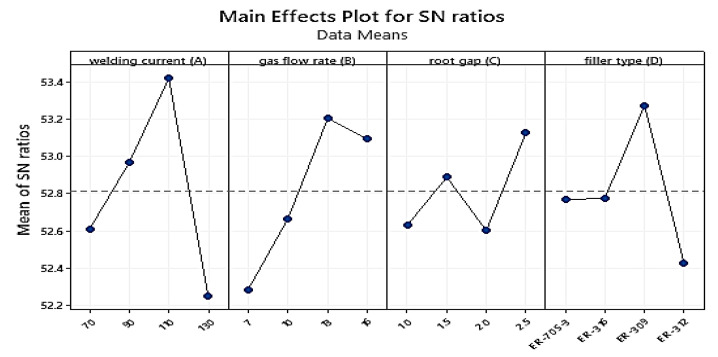
Main effect plot for S/N ratios for Tensile Strength.

**Figure 20 materials-15-04426-f020:**
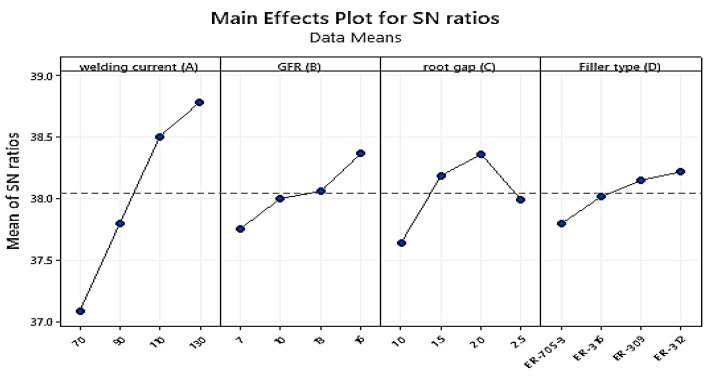
Main effect plot for signal-to-noise ratios for Hardness.

**Figure 21 materials-15-04426-f021:**
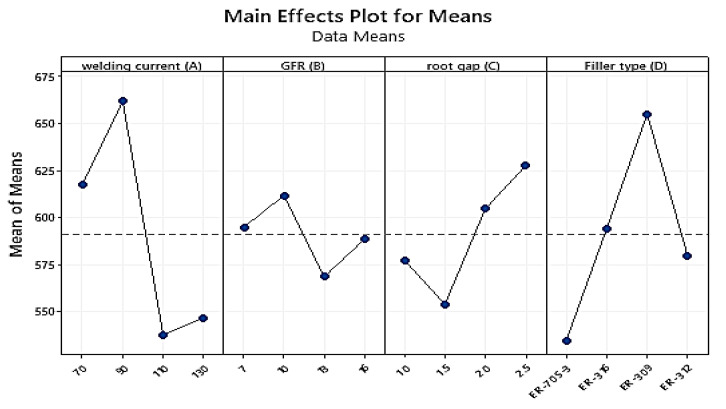
Main effect plot for the signal-to-noise ratios for Flexural Strengths.

**Figure 22 materials-15-04426-f022:**
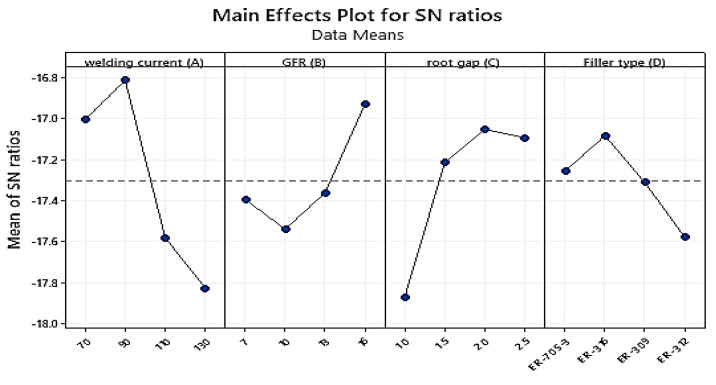
Main effect plot for S/N ratios of Bead Widths.

**Figure 23 materials-15-04426-f023:**
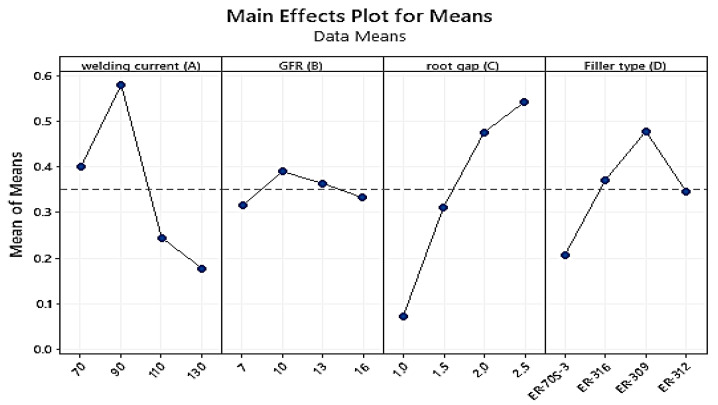
Main effect plot for means of CDI (dg).

**Table 1 materials-15-04426-t001:** Chemical composition of studied base and weld metals.

Materials (AISI Designation)		Chemical Composition (wt.%)									
		C	Mn	P	S	Si	Cr	Ni	Mo	N	Fe
Base metals	AISI 316	0.08	1.95	0.045	0.03	0.70	16.50	12.06	2.40	0.10	65.180
	AISI1020	0.20	0.40	0.042	0.03	-	-	-	-	-	99.328
Weld metal	Filler used										
	ER-316	0.063	1.70	0.043	0.03	0.40	14.61	10.74	1.90	0.078	-
	ER-309	0.063	1.75	0.041	0.03	0.67	17.99	9.81	0.89	0.015	0.52
	ER-312	0.119	1.72	0.034	0.03	0.46	20.25	11.05	0.89	0.015	0.52
	ER-70S-3	0.112	1.03	0.027	0.03	0.42	2.843	1.89	0.36	0.015	0.25

**Table 2 materials-15-04426-t002:** Chemical composition of commercial filler metals.

Materials (AWS Designation)		Chemical Composition (wt.%)										
		C	Mn	P	S	Si	Cr	Ni	Mo	N	Cu	Fe
Filler metals	ER-316L	0.03	1.92	0.043	0.03	0.42	16.98	12.05	2.21	0.09	-	Bal.
	ER-309L	0.03	2.00	0.04	0.03	0.80	23.06	13.20	0.75	-	0.74	Bal.
	ER-312	0.11	1.95	0.03	0.03	0.50	25.00	10.00	0.75	-	0.74	Bal.
	ER-70S-3	0.10	0.98	0.021	0.03	0.46	0.14	0.12	-	-	0.35	Bal.

Shielding gas used: pure argon; electrode used: non-consumable tungsten with 2% thorium electrode and a 2.4 mm diameter.

**Table 3 materials-15-04426-t003:** Selected process parameters and their levels.

S. No	Process Parameter	Units	Levels			
			1	2	3	4
1	Welding current	Amp (A)	70	90	110	130
2	Root Gap	Mm	1	1.5	2	2.5
3	Gas flow rate	L/Min	7	10	13	16
4	Type of filler metals	Type	ER316	ER309	ER70S-3	ER312

**Table 4 materials-15-04426-t004:** Experimental plan for similar and dissimilar metals, L16 OA.

S. No	Welding Current (A)	Gas Flow Rate (L/min)	Root Gap (mm)	Filler Type
S1	70	7	1	ER70S-3
S2	70	10	1.5	ER316
S3	70	13	2	ER309
S4	70	16	2.5	ER312
S5	90	7	1.5	ER309
S6	90	10	1	ER312
S7	90	13	2.5	ER70S-3
S8	90	16	2	ER316
S9	110	7	2	ER312
S10	110	10	2.5	ER309
S11	110	13	1	ER316
S12	110	16	1.5	ER70S-3
S13	130	7	2.5	ER316
S14	130	10	2	ER70S-3
S15	130	13	1.5	ER312
S16	130	16	1	ER309

**Table 5 materials-15-04426-t005:** Liquid penetrant testing result for similar TIG-welded samples.

Sample No	Non-Destructive Test Result
Sample 1	Undercut
Sample 2	Lack of fusion
Sample 3	Defect-free
Sample 4	Defect-free
Sample 5	Lack of fusion
Sample 6	Defect-free
Sample 7	Defect-free
Sample 8	Defect-free
Sample 9	Defect-free
Sample 10	Defect-free
Sample 11	Defect-free
Sample 12	Defect-free
Sample 13	Porosity
Sample 14	Defect-free
Sample 15	Defect-free
Sample 16	Defect-free

**Table 6 materials-15-04426-t006:** Process parameters and levels selected.

S. No	Process Parameter	Units	Levels			
			1	2	3	4
1	Welding current (A)	Amp (A)	70	90	110	130
2	Gas flow rate (B)	L/Min	7	10	13	16
3	Root gap (C)	mm	1	1.5	2	2.5
4	Type of filler metals (D)	Grade	ER-70S-3	ER-316	ER-309	ER-312

**Table 7 materials-15-04426-t007:** Mean tensile strength result.

Sample No	Welding Current (A)	Gas Flow Rate (B)	Root Gap (C)	Filler Type (D)	Tensile Strength (MPa)	Signal-to-Noise Ratio
1	70	7	1.0	ER70S-3	390	51.8213
2	70	10	1.5	ER316	420	52.4650
3	70	13	2.0	ER309	460	53.2552
4	70	16	2.5	ER312	442	52.9084
5	90	7	1.5	ER309	450	53.0643
6	90	10	1.0	ER312	410	52.2557
7	90	13	2.5	ER70S-3	478	53.5886
8	90	16	2.0	ER316	445	52.9672
9	110	7	2.0	ER312	410	52.2557
10	110	10	2.5	ER309	502	54.0141
11	110	13	1.0	ER316	483	53.6789
12	110	16	1.5	ER70S-3	486	53.7327
13	130	7	2.5	ER316	398	51.9977
14	130	10	2.0	ER70S-3	395	51.9319
15	130	13	1.5	ER312	412	52.2979
16	130	16	1.0	ER309	435	52.7698

**Table 8 materials-15-04426-t008:** Response table for Tensile Strength.

Level	Welding Current (A)	Gas Flow Rate (B)	Root Gap (C)	Filler Type (D)
1	428.0	412.0	429.5	437.3
2	445.8	431.8	442.0	436.5
3	**470.3**	**458.3**	427.5	**461.8**
4	410.0	452.0	**455.0**	418.5
Delta	60.3	46.3	27.5	43.3
Rank	1	2	4	3

**Table 9 materials-15-04426-t009:** Response table for signal-to-noise ratios for Tensile Strength (larger is better).

Level	Welding Current (A)	Gas Flow Rate (B)	Root Gap (C)	Filler Type (D)
1	52.61	52.28	52.63	52.77
2	52.97	52.67	52.89	52.78
3	53.42	53.21	52.60	53.28
4	52.25	53.09	53.13	52.43
Delta	1.17	0.92	0.52	0.85
**Rank**	**1**	**2**	**4**	**3**

**Table 10 materials-15-04426-t010:** Analysis of variance (ANOVA) for Tensile Strength.

Source	DF	Seq SS	% Cont.	Adj SS	Adj MS	F	P
Welding current (A)	3	3.00449	40.80	3.00449	1.00150	73.76	0.003
Gas flow rate (B)	3	2.13415	29.00	2.13415	0.71138	52.39	0.004
Root gap (C)	3	0.72768	9.90	0.72768	0.24256	17.86	0.020
Filler type (D)	3	1.45828	19.80	1.45828	0.48609	35.80	0.008
Residual error	3	0.04073	0.50	0.04073	0.01358		
Total	15	7.36534	100.00				

S = 0.0953%, R-Sq = 99.84%, R-Sq (adj) = 99.20%.

**Table 11 materials-15-04426-t011:** Mean hardness testing result.

Sample No	Welding Current (A)	Gas Flow Rate (B)	Root Gap (C)	Filler Type (D)	Hardness (HRB)	Signal-to-Noise Ratio
1	70	7	1.0	ER70S-3	64	36.1236
2	70	10	1.5	ER316	72	37.1466
3	70	13	2.0	ER309	75	37.5012
4	70	16	2.5	ER312	76	37.6163
5	90	7	1.5	ER309	78	37.8419
6	90	10	1.0	ER312	75	37.5012
7	90	13	2.5	ER70S-3	75	37.5012
8	90	16	2.0	ER316	83	38.3816
9	110	7	2.0	ER312	86	38.6900
10	110	10	2.5	ER309	84	38.4856
11	110	13	1.0	ER316	81	38.1697
12	110	16	1.5	ER70S-3	86	38.6900
13	130	7	2.5	ER316	83	38.3816
14	130	10	2.0	ER70S-3	88	38.8897
15	130	13	1.5	ER312	90	39.0849
16	130	16	1.0	ER309	87	38.7904

**Table 12 materials-15-04426-t012:** Response table for means of Hardness.

Level	Welding Current (A)	Gas Flow Rate (B)	Root Gap (C)	Filler Type (D)
1	71.75	77.75	76.75	78.25
2	77.75	79.75	81.50	79.75
3	84.25	80.25	83.00	81.00
4	87.00	83.00	79.50	81.75
Delta	15.25	5.25	6.25	3.50
Rank	1	3	2	4

**Table 13 materials-15-04426-t013:** Response table for signal-to-noise ratios for Hardness (larger is better).

Level	Welding Current (A)	Gas Flow Rate (B)	Root Gap (C)	Filler Type (D)
1	37.24	**38.43**	38.42	37.71
2	38.54	38.32	37.89	**39.02**
3	38.57	38.10	**38.54**	38.35
4	**38.74**	38.24	38.24	38.01
Delta	1.50	0.34	0.65	1.30
Rank	1	4	3	2

**Table 14 materials-15-04426-t014:** Analysis of variance (ANOVA) for Hardness.

Source	DF	Seq SS	% Cont.	Adj SS	Adj MS	F	P
Welding current (A)	3	6.88294	74.57	6.88294	2.29431	200.49	0.001
GFR (B)	3	0.75520	8.52	0.75520	0.25173	22.00	0.015
Root gap (C)	3	1.14149	12.37	1.14149	0.38050	33.25	0.008
Filler type (D)	3	0.41515	4.50	0.41515	0.13838	12.09	0.035
Residual error	3	0.03433	0.04	0.03433	0.01144		
Total	15	9.22910	100				

S = 0.2276, R-Sq = 98.57%, R-Sq (adj) = 92.86%.

**Table 15 materials-15-04426-t015:** Flexural strength test result.

Sample No	Welding Current (A)	Gas Flow Rate (B)	Root Gap (C)	Filler Type (D)	Flexural Strength (MPa)	Signal-to-Noise Ratio
1	70	7	1.0	ER703	555	54.8859
2	70	10	1.5	ER316	605	55.6351
3	70	13	2.0	ER309	670	56.5215
4	70	16	2.5	ER312	640	56.1236
5	90	7	1.5	ER309	692	56.8021
6	90	10	1.0	ER312	655	56.3248
7	90	13	2.5	ER703	620	55.8478
8	90	16	2.0	ER316	679	56.6374
9	110	7	2.0	ER312	545	54.7279
10	110	10	2.5	ER309	662	56.4172
11	110	13	1.0	ER316	505	54.0658
12	110	16	1.5	ER703	440	52.8691
13	130	7	2.5	ER316	588	55.3875
14	130	10	2.0	ER703	525	54.4032
15	130	13	1.5	ER312	480	53.6248
16	130	16	1.0	ER309	595	55.4903

**Table 16 materials-15-04426-t016:** Response table for means of Flexural Strengths.

Level	Welding Current (A)	Gas Flow Rate (B)	Root Gap (C)	Filler Type (D)
1	617.5	595.0	577.5	535.0
2	661.5	611.8	554.3	594.3
3	538.0	568.8	604.8	654.8
4	547.0	588.5	627.5	580.0
Delta	123.5	43.0	73.3	119.8
Rank	1	4	3	2

**Table 17 materials-15-04426-t017:** Response table for signal-to-noise ratios of Flexural Strengths (larger is better).

Level	Welding Current (A)	Gas Flow Rate (B)	Root Gap (C)	Filler Type (D)
1	55.85	55.45	55.09	54.42
2	56.36	55.30	54.99	55.47
3	54.37	55.31	55.61	56.28
4	54.73	55.23	55.61	55.13
Delta	1.99	0.22	0.62	1.86
Rank	1	4	3	2

**Table 18 materials-15-04426-t018:** Analysis of variance (ANOVA) for Flexural Strengths.

Source	DF	Seq SS	% Cont.	Adj SS	Adj MS	F	P
Welding current (A)	3	9.5245	46.57	9.52446	3.17482	190.89	0.001
GFR (B)	3	0.9838	4.8	0.98376	0.32792	19.72	0.018
Root gap (C)	3	3.2319	15.80	3.23193	1.07731	64.77	0.003
Filler type (D)	3	6.6638	32.58	6.66380	2.22127	133.55	0.001
Residual error	3	0.0499	0.25	0.04990	0.01663		
Total	15	20.4538	100				

S = 0.2597, R-Sq = 98.64%, R-Sq (adj) = 93.19%.

**Table 19 materials-15-04426-t019:** The experimental results of Bead width.

Sample No	Welding Current (A)	Gas Flow Rate (B)	Root Gap (C)	Filler Type (D)	Bead Width (mm)	Signal-to-Noise Ratio
1	70	7	1.0	ER703	7.50	−17.5012
2	70	10	1.5	ER316	7.05	−16.9638
3	70	13	2.0	ER309	6.90	−16.7770
4	70	16	2.5	ER312	6.90	−16.7770
5	90	7	1.5	ER309	7.00	−16.9020
6	90	10	1.0	ER312	7.80	−17.8419
7	90	13	2.5	ER703	6.80	−16.6502
8	90	16	2.0	ER316	6.21	−15.8618
9	110	7	2.0	ER312	7.70	−17.7298
10	110	10	2.5	ER309	7.50	−17.5012
11	110	13	1.0	ER316	8.00	−18.0618
12	110	16	1.5	ER703	7.10	−17.0252
13	130	7	2.5	ER316	7.45	−17.4431
14	130	10	2.0	ER703	7.80	−17.8419
15	130	13	1.5	ER312	7.90	−17.9525
16	130	16	1.0	ER309	8.00	−18.0618

**Table 20 materials-15-04426-t020:** Response table for means of Bead Widths.

Level	Welding Current (A)	Gas Flow Rate (B)	Root Gap (C)	Filler Type (D)
1	7.088	7.412	7.825	7.300
2	6.953	7.538	7.262	7.177
3	7.575	7.400	7.152	7.350
4	7.787	7.053	7.162	7.575
Delta	0.835	0.485	0.673	0.397
Rank	1	3	2	4

**Table 21 materials-15-04426-t021:** Response table for signal-to-noise ratios of Bead Widths (smaller is better).

Level	Welding Current (A)	Gas Flow Rate (B)	Root Gap (C)	Filler Type (D)
1	−17.00	−17.39	−17.87	−17.25
2	−16.81	−17.54	−17.21	−17.08
3	−17.58	−17.36	−17.05	−17.31
4	−17.82	−16.93	−17.09	−17.58
Delta	1.01	0.61	0.81	0.49
Rank	1	3	2	4

**Table 22 materials-15-04426-t022:** Analysis of variance (ANOVA) of Bead Widths.

Source	DF	Seq SS	% Cont.	Adj SS	Adj MS	F	P
Welding current (A)	3	2.70740	46.35	2.70740	0.90247	31.29	0.009
GFR (B)	3	0.81780	14.00	0.81780	0.27260	9.45	0.049
Root gap (C)	3	1.73212	29.64	1.73212	0.57737	20.02	0.017
Filler type (D)	3	0.50032	8.56	0.50032	0.16677	5.78	0.092
Residual error	3	0.08652	1.45	0.08652	0.02884		
Total	15	5.84416	100				

S = 0.1008, R-Sq = 99.54%, R-Sq (adj) = 97.70%.

**Table 23 materials-15-04426-t023:** Result table for all responses.

Sample No	Welding Current (A)	Gas Flow Rate (B)	Root Gap (C)	Filler Type (D)	Tensile Strength (MPa)	Hardness (HRB)	Flexural Strength (MPa)	Bead Width (mm)
1	70	7	1.0	ER70S-3	390	64	555	7.50
2	70	10	1.5	ER316	420	72	605	7.05
3	70	13	2.0	ER309	460	75	670	6.90
4	70	16	2.5	ER312	442	76	640	6.90
5	90	7	1.5	ER309	450	78	692	7.00
6	90	10	1.0	ER312	410	75	655	7.80
7	90	13	2.5	ER70S-3	478	75	620	6.80
8	90	16	2.0	ER316	445	83	679	6.21
9	110	7	2.0	ER312	410	86	545	7.70
10	110	10	2.5	ER309	502	84	662	7.50
11	110	13	1.0	ER316	483	81	505	8.00
12	110	16	1.5	ER70S-3	486	86	440	7.10
13	130	7	2.5	ER316	398	83	588	7.45
14	130	10	2.0	ER70S-3	395	88	525	7.80
15	130	13	1.5	ER312	412	90	480	7.90
16	130	16	1.0	ER309	435	87	595	8.00

**Table 24 materials-15-04426-t024:** Individual desirability index and composite desirability index with its rank.

UTS	HARD	FLEX.STR	BW	Individual Desirability Index (di)				CDI (dg)	Rank
MPa	HRB	MPa	mm	UTS	Hardness	Flexural	Bead Width		
390	64	555	7.5	0.0000	0.0000	0.6755	0.5285	0.0000	13
420	72	605	7.05	0.5175	0.5547	0.8092	0.7285	0.4114	7
460	75	670	6.9	0.7906	0.6504	0.9554	0.7839	0.6206	5
442	76	640	6.9	0.6814	0.6794	0.8909	0.7839	0.5686	6
450	78	692	7	0.7319	0.7338	1.0000	0.7474	0.6336	3
410	75	655	7.8	0.4226	0.6504	0.9237	0.3343	0.2913	10
478	75	620	6.8	0.8864	0.6504	0.8452	0.8188	0.6316	4
445	83	679	6.21	0.7008	0.8549	0.9739	1.0000	0.7638	1
410	86	545	7.7	0.4226	0.9199	0.6455	0.4094	0.3205	8
502	84	662	7.5	1.0000	0.8771	0.9386	0.5285	0.6596	2
483	81	505	8	0.9112	0.8086	0.5079	0.0000	0.0000	13
486	86	440	7.1	0.9258	0.9199	0.0000	0.7091	0.0000	13
398	83	588	7.45	0.2673	0.8549	0.7664	0.5543	0.3115	9
395	88	525	7.8	0.2113	0.9608	0.5808	0.3343	0.1985	12
412	90	480	7.9	0.4432	1.0000	0.3984	0.2364	0.2043	11
435	87	595	8	0.6339	0.9405	0.7843	0.0000	0.0000	13

**Table 25 materials-15-04426-t025:** Initial parametric setting.

Initial Setting	UTS (MPa)	HARDNESS (HRB)	FS (MPa)	BW (mm)	CDI (dg)
A2 B4 C3 D2	445	83	679	6.21	0.7638

**Table 26 materials-15-04426-t026:** Response table for means (CDI).

Level	Welding Current (A)	Gas Flow Rate (B)	Root Gap (C)	Filler Type (D)
1	0.40013	0.31641	0.07283	0.20754
2	0.58009	0.39020	0.31232	0.37168
3	0.24503	0.36413	0.47585	0.47844
4	0.17859	0.33310	0.54284	0.34617
Delta	0.40150	0.07380	0.47001	0.27090
Rank	2	4	1	3

**Table 27 materials-15-04426-t027:** Analysis of variance of dg.

Source	DF	Seq SS	% Cont.	Adj SS	Adj MS	F	P
welding current (A)	3	0.38341	35	0.38341	0.127803	18.19	0.020
GFR (B)	3	0.01291	1.20	0.01291	0.004302	0.61	0.652
root gap (C)	3	0.52506	48	0.52506	0.175020	24.91	0.013
Filler type (D)	3	0.14909	14	0.14909	0.049697	7.07	0.071
Residual Error	3	0.02107	1.80	0.02107	0.007025		
Total	15	1.09154	100				

S = 0.05481, R-Sq = 98.42%, R-Sq (adj) = 92.12%.

**Table 28 materials-15-04426-t028:** The optimum parametric setting.

Optimum Setting	UTS (MPa)	HARDNESS (HRB)	FS (MPa)	BW (mm)	CDI (dg)
A2 B2 C4 D3	493.2	86	691	5.8	0.9387

**Table 29 materials-15-04426-t029:** Improvement in UTS, FS, HARDNESS, and BW.

Initial Setting				Optimum Setting				% Improvement			
A2 B4 C3 D2				A2 B2 C4 D3							
UTS	HRD	FS	BW	UTS	HRD	FS	BW	UTS	HRD	FS	BW
445	83	679	6.21	493.2	86	691	5.8	10.83	3.61	1.77	6.60
**Composite desirability**											
0.7638				0.9387				22.90			

UTS: ultimate tensile strength; HRD: hardness; FS: flexural strength; BW: bead width.

**Table 30 materials-15-04426-t030:** Confirmation test.

Type of Samples	Optimum Setting	UTS (MPa)	HARDNESS (HRB)	FS (MPa)	BW (mm)	CDI (dg)	Error
**Dissimilar**	A2 B2 C4 D3	493.2	86	691	5.80	0.9387	**3.42%**
	**Exp. setting**	498	88	691.45	5.25	0.9708	
	A2 B2 C4 D3						

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
