# Peer review of "Experimental Investigation and Parametric Optimization of the Tungsten Inert Gas Welding Process Parameters of Dissimilar Metals"

_materials, 2022, doi:10.3390/ma15134426_

Round 1
Reviewer 1 Report
This paper studied the tungsten inert gas welding of dissimilar (SS316 with AISI 1020) steels. It must be revised before it can be accepted for publication:
Abbreviation should be explained before use. TIG
The abstract has too many words. It needs to be condensed. e.g. ‘These two metals were commonly used in industrial applications such as automotive engineering, biomedical engineering, dairy industry, and general mechanical engineering products’ this is background information and should be put in the introduction. ‘As a result, the above TIG welding parameters for tensile strength, hardness, flexural strength, and bead width of welds was investigated and optimized’ this can be combined with its previous sentence…; ‘Universal testing machine was used to test tensile and flexural strength of welded metals. Rockwell and Vickers hardness testing machine was used to justify the hardness’ this is not quite important and necessary information to be put here. Anyway, it needs to be significantly condensed, only show the main methods, results and conclusions in the abstract.
GTAW (gas tungsten arc welding), this should be put the other way around. Abbreviation should be in the bracket.
Line 103, ‘it was .0discovered’?
The in-text reference format is not proper, either use ‘author+year’ or ‘author [number]’, but not the combination of them. For this journal <Materials>, it is the format of ‘author[number]’. ‘[12] investigated…’ where are the author names before [12]?
In introduction, the importance of TIG over the conventional mechanical joining techniques should be emphasized based on the potential drawbacks of the latter. See: high stress concentration (10.1016/j.compositesb.2015.07.018) and uneven load distribution and low fatigue strength (10.1016/j.compstruct.2015.08.113) for conventional mechanical joining, and revise accordingly.
The literature review section needs to be properly sectioned, dividing it into several paragraphs, not all in just one paragraph. Add the latest review for TIG welding: 10.1016/j.matpr.2022.04.266
The novelty of this work should be highlighted at the end of the literature review section, what’s novelty compared with previous work?
Figure caption is too short, e.g Fig. 2. Actually, this figure is not quite necessary as it doesn’t show any more information except the apperance.
Figure caption could be revised for Fig. 3. It doesn’t show the Chemical composition. Perhaps ‘Experimental set-up of …’ is more proper.
Table 1 and 2 should be combined into one Table.
Combine Figs5-6 into one figure, revise the figure caption. e.g. fig.5b the caption is missing
Combine Figs7-8 into one figure, revise the figure caption.
FIg. 9 needs more details, either add notes in the figure, or add more detailed figure caption. What do different figures represent?
Fix the issues in Fig 13 caption. It has lots of garbled words.
Combine Figs14-16 into one figure, using subfigures.
Fig. 17 is not necessary. Fig. 19 is not necessary. Just show the locations where the hardness tests are conducted are enough.
Combine Figs20-21 into one figure.
Fig. 23 is not necessary. Fig. 24 is not necessary. Fig. 25 is not necessary These are basic knowledge and standard procedures. As commented before, Just show the locations where the hardness/Optical/SEM tests are conducted are enough. These procedures can be briefly mentioned in the text, that would be enough.
There are just too many figures, the authors should know that a scientific paper only shows the most important results, not all the basic knowledge/results, that will make it like a report or thesis, not a scientific paper. A paper should be concise.
Author Response
Title: Experimental Investigation and Parametric Optimization of Tungsten Inert Gas Welding Process Parameters of Dissimilar Metals
MATERIALS-ID-1758917
The authors are highly grateful to the reviewers for their constructive comments, which have helped to enhance the quality of the manuscript. Author's sincere effort has been put to revise the manuscript according to the reviewer comments. The details of revisions made in response to the comments are summarized in the following Table. The revisions in the manuscript are shown in YELLOW colored texts.
|
|
REVIEWER-COMMENT-1 |
AUTHOR RESPONSE |
1.The abstract has too many words. It needs to be condensed. e.g. ‘These two metals were commonly used in industrial applications such as automotive engineering, biomedical engineering, dairy industry, and general mechanical engineering products’ this is background information and should be put in the introduction. ‘As a result, the above TIG welding parameters for tensile strength, hardness, flexural strength, and bead width of welds was investigated and optimized’ this can be combined with its previous sentence…; ‘Universal testing machine was used to test tensile and flexural strength of welded metals. Rockwell and Vickers hardness testing machine were used to justify the hardness’ this is not quite important and necessary information to be put here. Anyway, it needs to be significantly condensed, only show the main methods, results and conclusions in the abstract. GTAW (gas tungsten arc welding) |
Reply: As per the Reviewer suggestion we have minimized the number of words in the abstract and it is around 252, and has been significantly condensed and concentrated only on methods, results and conclusions in the abstract. |
2. GTAW (gas tungsten arc welding), this should be put the other way around. Abbreviation should be in the bracket. |
Reply: As per the Reviewers comment it has been corrected, On line number 57. |
3. Line 103, ‘it was .0discovered’? |
Reply: It was a typing error and has been corrected, On line number 103. |
4. The in-text reference format is not proper, either use ‘author+year’ or ‘author [number]’, but not the combination of them. For this journal, it is the format of ‘author[number]’. ‘[12] investigated…’ where are the author names before [12]? |
Reply: As per the Reviewers comment it has been corrected, in the Literature Section as ‘author[number]’. |
5. In introduction, the importance of TIG over the conventional mechanical joining techniques should be emphasized based on the potential drawbacks of the latter. See: high stress concentration (10.1016/j.compositesb.2015.07.018) and uneven load distribution and low fatigue strength (10.1016/j.compstruct.2015.08.113) for conventional mechanical joining, and revise accordingly. |
Reply: As per the Reviewers comment, related content of TIG has been added to emphasis on the importance and it has been colored YELLOW in the text on Line Number |
6.The literature review section needs to be properly sectioned, dividing it into several paragraphs, not all in just one paragraph. Add the latest review for TIG welding: 10.1016/j.matpr.2022.04.266 |
Reply: As per the Reviewers comment, the literature section has been divided into paragraphs and sub sectioned as suggested. |
7. The novelty of this work should be highlighted at the end of the literature review section, what’s novelty compared with previous work? |
Reply: As per the Reviewers comment, The Novelty of the work has been added at the end of literature and highlighted. Line Number |
8. Figure caption is too short, e.g Fig. 2. Actually, this figure is not quite necessary as it doesn’t show any more information except the appearance. |
Reply: As per the Reviewers comment, Figure 2: it has been removed from the text. Line Number |
9. Figure caption could be revised for Fig. 3. It doesn’t show the Chemical composition. Perhaps ‘Experimental set-up of …’ is more proper |
Reply: As per the Reviewers comment Figure Caption has been corrected to Experimental Setup for Spectrometry and it has been highlighted. Line Number |
10. Table 1 and 2 should be combined into one Table. |
Reply: As per the Reviewers comment Table 1 and 2 has been combined and it has been highlighted. Line Number |
11. Combine Figs5-6 into one figure, revise the figure caption. e.g. fig.5b the caption is missing |
Reply: As per the Reviewers comment Figure 5 & 6 has been combined and caption added. Line Number |
12. Combine Figs7-8 into one figure, revise the figure caption. |
Reply: As per the Reviewers comment Figure 7 & 8 has been combined and caption added. Line Number |
13. FIg. 9 needs more details, either add notes in the figure, or add more detailed figure caption. What do different figures represent? |
Reply: As per the Reviewers comment, In Figure 9 caption is added and Figure is corrected on Line Number |
14. Fix the issues in Fig 13 caption. It has lots of garbled words. |
Reply: We agree with the Reviewers comment, As ASTM standard was well known for Tensile testing Specimen and therefore the figure was removed from the text. |
15. Fig. 17 is not necessary. Fig. 19 is not necessary. Just show the locations where the hardness tests are conducted are enough. |
Reply: We agree with the Reviewers comment, it has been mentioned in the Text on Line Number and figure 17 & 19 were removed. |
16. Combine Figs20-21 into one figure. |
Reply: We agree with the Reviewers comment, Figs 20-21 has been modified as one Figure on Line Number as suggested. |
17. Fig. 23 is not necessary. Fig. 24 is not necessary. Fig. 25 is not necessary These are basic knowledge and standard procedures. As commented before, just show the locations where the hardness/Optical/SEM tests are conducted are enough. These procedures can be briefly mentioned in the text, that would be enough. |
Reply: We agree with the Reviewers comment, it has been mentioned in the Text on Line Number and figure 23, 24 & 25 were removed and mentioned in the text. |
18. There are just too many figures, the authors should know that a scientific paper only shows the most important results, not all the basic knowledge/results, that will make it like a report or thesis,  not a scientific paper. A paper should be concise. |
Reply: We agree with the Reviewers comment. We have reduced the figures in the manuscript from overall 43 Figures to 23 Figures in the Revised version of the Manuscript. And also the Number of Pages were also reduced from 39 page to 36 |
|

Reviewer 2 Report
This paper has stringent experiments and a very detailed discussion, but there are still big problems.
1. Innovation is doubtful, this paper is more like an experimental report or popular science article.
The authors spend nearly half of the paper to introduce the experimental equipment and operating principles, which is relatively engineering content. As an academic paper, it is not necessary to have a long introduction to these existing contents without own ideas.
In my opinion, The result and discussion section is a step response experiment. The experimental results are useless once the parameters, welding processes, and materials change slightly. Do we still need to do so much repetitive work?
The Most important: there are no core algorithms or ideas in this paper. The authors use existing methods to do analysis on a particular welding process, i.e., it is not innovative. It is not informative for other researchers.
2. Abstract is too long, reduce it to about 200 words.
3. The picture is stretched and distorted, not convenient to read.
Author Response
Title: Experimental Investigation and Parametric Optimization of Tungsten Inert Gas Welding Process Parameters of Dissimilar Metals
MATERIALS-ID-1758917
The authors are highly grateful to the reviewers for their constructive comments, which have helped to enhance the quality of the manuscript. Author's sincere effort has been put to revise the manuscript according to the reviewer comments. The details of revisions made in response to the comments are summarized in the following Table. The revisions in the manuscript are shown in YELLOW colored texts.
|
|
REVIEWER-COMMENT-2 |
AUTHOR RESPONSE |
1.Innovation is doubtful, this paper is more like an experimental report or popular science article
The authors spend nearly half of the paper to introduce the experimental equipment and operating principles, which is relatively engineering content. As an academic paper, it is not necessary to have a long introduction to these existing contents without own ideas
The Most important: there are no core algorithms or ideas in this paper. The authors use existing methods to do analysis on a particular welding process, i.e., it is not innovative. It is not informative for other researchers. |
Reply: We Agree with the Reviewer comment, our manuscript mainly concentrated on experimental work and investigation of Influential Parameters on TIG and the results were broadly discussed and conclusions are presented. A parametric Optimisation method has been utilized to determine the optimum parametric setting for good weld quality and checked by further validation of experimental results. |
2. Abstract is too long, reduce it to about 200 words. |
Reply: As per the Reviewer suggestion we have minimized the number of words in the abstract and it is around 252, and has been significantly condensed and concentrated only on methods, results and conclusions in the abstract. |
3. The picture is stretched and distorted, not convenient to read. |
Reply: We agree with the Reviewers comment. We have Increase the DPI and visibility of all the figures and also some of the figures which are not clarity has been removed and replaced by the text in the manuscript. The figures were reduced in the manuscript from overall 43 Figures to 23 Figures in the Revised version of the Manuscript. And also the Number of Pages were also reduced from 39 page to 36 |
|

Round 2
Reviewer 2 Report
no